# Kernel PCA for Out-of-Distribution Detection

**Kun Fang**[1]    **Qinghua Tao**[2]    **Kexin Lv**[3]    **Mingzhen He**[1]    **Xiaolin Huang**[1]    **Jie Yang**[1]

[1]Shanghai Jiao Tong University    [2]ESAT-STADIUS, KU Leuven
[3]China Mobile Shanghai ICT Co., Ltd

{fanghenshao,mingzhen_he,xiaolinhuang,jieyang}@sjtu.edu.cn
qinghua.tao@esat.kuleuven.be    lvkexin@cmsr.chinamobile.com

## Abstract

Out-of-Distribution (OoD) detection is vital for the reliability of Deep Neural Networks (DNNs). Existing works have shown the insufficiency of Principal Component Analysis (PCA) straightforwardly applied on the features of DNNs in detecting OoD data from In-Distribution (InD) data. The failure of PCA suggests that the network features residing in OoD and InD are not well separated by simply proceeding in a linear subspace, which instead can be resolved through proper non-linear mappings. In this work, we leverage the framework of Kernel PCA (KPCA) for OoD detection, and seek suitable non-linear kernels that advocate the separability between InD and OoD data in the subspace spanned by the principal components. Besides, explicit feature mappings induced from the devoted task-specific kernels are adopted so that the KPCA reconstruction error for new test samples can be efficiently obtained with large-scale data. Extensive theoretical and empirical results on multiple OoD data sets and network structures verify the superiority of our KPCA detector in efficiency and efficacy with state-of-the-art detection performance.

## 1   Introduction

With the rapid advancement of the powerful learning abilities of Deep Neural Networks (DNNs) [1, 2], the trustworthiness of DNNs in security-sensitive scenarios has attracted considerable attention in recent years [3]. Generally, samples from the training and test sets of DNNs are viewed as data from some In Distribution (InD) $\mathbb{P}_{in}$, while samples from other data sets are regarded as coming from a different distribution $\mathbb{P}_{out}$, *i.e.*, out-of-distribution (OoD) data. In the practical deployment, DNNs trained on InD data would inevitably encounter OoD data and thus yield unreliable results with potential risks. Therefore, detecting whether a new sample is from $\mathbb{P}_{in}$ or $\mathbb{P}_{out}$ has been a valuable research topic of trustworthy deep learning, namely OoD detection [4].

Existing OoD detection methods exploit different outputs from DNNs to unveil the hidden disparities between InD and OoD data, *e.g.*, logits [5], gradients [6] and features [7, 8]. In this work, we address OoD detection from a perspective of utilizing the feature spaces learned by the backbone of DNNs. To be specific, given a DNN $f : \mathbb{R}^d \to \mathbb{R}^c$, $f$ takes $x \in \mathbb{R}^d$ as inputs and learns penultimate layer features $z \in \mathbb{R}^m$ before the last linear layer. Principal Component Analysis (PCA) is investigated in [8] to calculate the reconstruction error in the $z$-space as the OoD detection score. That is, PCA is executed on the penultimate features of InD training samples and learns a linear subspace spanned by the principal components in the $z$-space. Then, given features $\hat{z}$ of an unknown sample $\hat{x}$, one can obtain the reconstructed counterpart of $\hat{z}$ by projecting $\hat{z}$ to the linear subspace and re-projecting it back. The reconstruction error is computed as the Euclidean distance between $\hat{z}$ and the reconstructed counterpart. In [9], similar ideas are adopted with energy-based models using an

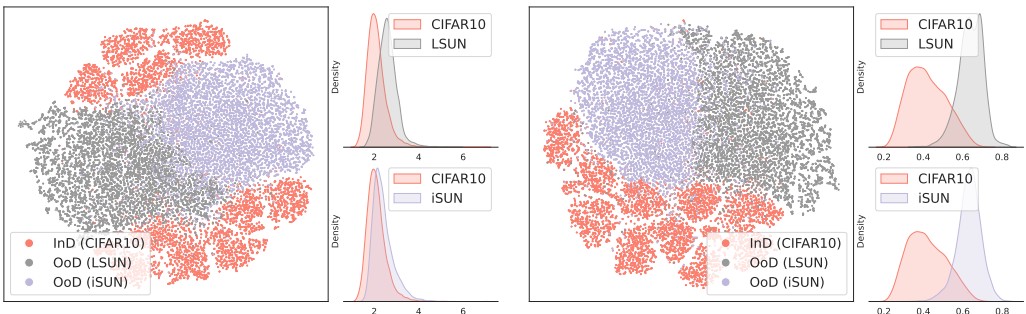

(a) T-SNE of $z$ and PCA reconstruction errors.  (b) T-SNE of $\Phi(z)$ and KPCA reconstruction errors.

Figure 1: The t-SNE [10] visualization on the original features $z$ (left) and the mapped features $\Phi(z)$ (right). Our KPCA detector alleviates the linearly inseparability between InD and OoD features in the original $z$-space via an explicit feature mapping $\Phi$, and thus substantially improves the OoD detection performance, illustrated by the much more distinguishable reconstruction errors.

auto-encoder structure, where the neural networks are trained from scratch and the reconstruction is conducted in the decoder. For good OoD detection performance, it is expected that InD features are compactly allocated along the linear principal components with high variances for capturing most of the informative patterns of InD data, leading to small reconstruction errors, while OoD features are not supposed to be well matched with the learned subspace, causing large reconstruction errors.

However, it has been empirically observed in [8] that such PCA reconstruction errors alone cannot distinctively differentiate OoD data from InD data, leading to poor detection performance of PCA in the $z$-space. Nevertheless, [8] did not take further explorations on the reasons behind, and instead proposed a practical fusion trick to boost PCA by multiplying with other existing powerful detection scores. Therefore, in this work, more in-depth analyses are undertaken to improve the limitations of PCA for OoD detection with insightful understandings on the distribution of InD and OoD features. It is widely acknowledged that PCA falls inferior in dealing with those linearly-inseparable data, which inspires us to explore the non-linearity existing in the $z$-space of InD and OoD features under the help of the celebrated Kernel PCA (KPCA).

KPCA has long been a powerful technique in learning the non-linear patterns of data [11, 12]. By deploying KPCA, a non-linear feature mapping $\Phi$ is imposed on the $z$-space in this setup, so that the linear inseparability can be alleviated in the mapped $\Phi(z)$-space. KPCA is generally conducted through a kernel function $k$ induced by the feature mapping, *i.e.*, $k(z_1, z_2) = \langle \Phi(z_1), \Phi(z_2) \rangle$, in order to avoid explicit calculations in the mapped $\Phi(z)$-space. In most cases, researchers have no prior on the unknown non-linear data distribution, *e.g.*, the non-linearity in the $z$-space of InD and OoD features. Therefore, finding an appropriate $k$ or $\Phi$ that well adapts the data always remains a non-trivial issue for KPCA. For example, the mostly common Gaussian kernel is shown to be unable to separate the InD and OoD features, and leads to terrible OoD detection performance, see details in Appendix C. In addition, KPCA also faces the challenge of calculating and storing the $N_{tr} \times N_{tr}$ kernel matrix on millions of training samples with a very large size $N_{tr}$, which significantly hinders its application in practical tasks with a huge amount of data.

The proposed KPCA detection method well addresses the aforementioned issues of KPCA for OoD detection. On the one hand, to better understand the non-linear patterns in InD and OoD features, we take a kernel perspective on an existing OoD detector [7], which searches $k$-th Nearest Neighbors (KNN) on the $\ell_2$ normalized features $z$. By decoupling and analyzing key components of the KNN method, we acquire two effective kernels, a cosine kernel and a cosine-Gaussian kernel, for our KPCA detector to promote the linear separability between InD and OoD features in the subspace of the principal components, leading to substantially improved distinguishable KPCA reconstruction errors, as shown in Figure 1. On the other hand, for a computationally-friendly implementation, two explicit feature mappings $\Phi$ induced from the cosine and cosine-Gaussian kernels are executed on the original features $z$, followed by PCA in the *mapped $\Phi(z)$-space* to obtain the reconstruction errors without calculations on the kernel matrix. Specifically, the celebrated Random Fourier Features (RFFs) [13] are introduced to approximate the Gaussian kernel, allowing an $\mathcal{O}(M)$ computation

complexity in inference, which significantly outperforms the $\mathcal{O}(N_{\mathrm{tr}})$ computation complexity of the KNN [7] and the kernel-matrix-based KPCA ($M$ is the number of RFFs and $M \ll N_{\mathrm{tr}}$).

Extensive experiments verify the effectiveness of the devoted two kernels, which we hope could bring inspirations for the research community in exploring the non-linearity in InD and OoD data from a kernel perspective. For example, the two kernels can even serve as a beneficial prior on advocating learning more and stronger kernels for OoD detection. In addition, we supplement our method with its implementation via the kernel matrix, and illustrate the advantageous effectiveness and efficiency of explicit feature mappings in Section 6. The contributions of this work are summarized below:

- To the best of our knowledge, this is the first work that explores suitable kernels to seize the non-linearity in InD and OoD features in the post-hoc stage on well-trained DNNs.

- Two *task-specific* kernels are carefully devised for OoD detection. Particularly, two explicit feature mappings induced from the kernels are adopted for the KPCA detector, and lead to separable KPCA reconstruction errors with significantly-reduced complexity in inference.

- Theoretical and experimental comparisons indicate the effects of kernels in our KPCA detector with SOTA detection performance and remarkably reduced time complexity.

In the remainder of this work, related works and research backgrounds are outlined in Section 2 and Section 3, respectively. Section 4 delves into details of the proposed KPCA detector. Comparison experiments with prevailing OoD detection methods and KPCA via kernel functions are presented in Section 5 and Section 6, respectively. Conclusions and limitations are drawn in Section 7.

## 2 Related work

Generally, out-of-distribution detection has been formulated as a binary classification problem including a decision function $D(\cdot)$ and a scoring function $S(\cdot)$:

$$D(\boldsymbol{x}) = \left\{ \begin{array}{ll} \mathrm{InD}, & S(\boldsymbol{x}) > s, \\ \mathrm{OoD}, & S(\boldsymbol{x}) < s. \end{array} \right. \tag{1}$$

The scoring function $S(\cdot)$ assigns a score $S(\hat{\boldsymbol{x}})$ for a new sample $\hat{\boldsymbol{x}}$. If $S(\hat{\boldsymbol{x}})$ is greater than a threshold $s$, the decision function $D(\cdot)$ would view $\hat{\boldsymbol{x}}$ as an InD sample, and vice versa. The key to effectually detecting OoD samples is a well-designed scoring function. Existing OoD detectors adopt different outputs from DNNs and design justified scores to measure the disparity between InD and OoD data.

**Logits-based** detectors exploit the abnormal responses reflected in the predictive logits or probabilities from DNNs to detect OoD data. Typical methods adopt either the maximum logits [14] or probability [15, 16] or the energy function on logits [5] as the detection score.

**Gradients-based** methods investigate differences on gradients *w.r.t* InD and OoD data for OoD detection. For example, gradient norms [6] or low-dimensional representations [17] are studied to devise the detection score.

**Features-based** detectors try to capture the feature information causing over-confidence of OoD predictions in different ways. Feature clipping [18, 19, 20, 21, 22], feature distances [23, 7, 24], feature norms [25], rank-1 features [26], feature subspace [8], etc., have been explored with excellent performance.

Aside from methods above, other existing OoD detectors cover the training regularization [27, 28], the ensemble technique [29] and theoretical understandings [30, 31]. Refer to Appendix A for more detailed descriptions on the compared methods in experiments [32, 33, 34, 35].

## 3 Background

### 3.1 PCA for OoD detection

The PCA detector with the reconstruction error as the detection score is summarized firstly. Given the penultimate features $\boldsymbol{z}_i \in \mathbb{R}^m$ learned by a well-trained DNN $f : \mathbb{R}^d \to \mathbb{R}^c$ of the InD training

data $\boldsymbol{x}_i \in \mathbb{R}^d, i = 1, \cdots, N_{\mathrm{tr}}$, the covariance matrix $\boldsymbol{\Sigma}$ is calculated as:

$$\boldsymbol{\Sigma} = \sum_{i=1}^{N_{\mathrm{tr}}} (\boldsymbol{z}_i - \boldsymbol{\mu}) (\boldsymbol{z}_i - \boldsymbol{\mu})^\top, \quad \boldsymbol{\mu} = \frac{1}{N_{\mathrm{tr}}} \sum_{i=1}^{N_{\mathrm{tr}}} \boldsymbol{z}_i. \tag{2}$$

Through the eigendecomposition $\boldsymbol{\Sigma} = \boldsymbol{U}\boldsymbol{\Lambda}\boldsymbol{U}^\top$, the dimensionality reduction matrix $\boldsymbol{U}_q \in \mathbb{R}^{m \times q}$ is obtained by taking the first $q$ columns of the eigenvector matrix $\boldsymbol{U}$ *w.r.t* the top-$q$ largest eigenvalues.

In inference, given a new sample $\hat{\boldsymbol{x}} \in \mathbb{R}^d$ and its feature $\hat{\boldsymbol{z}} \in \mathbb{R}^m$ from the DNN $f$, the reconstruction error is computed as:

$$e(\hat{\boldsymbol{x}}) = \left\| \boldsymbol{U}_q \boldsymbol{U}_q^\top (\hat{\boldsymbol{z}} - \boldsymbol{\mu}) - (\hat{\boldsymbol{z}} - \boldsymbol{\mu}) \right\|_2. \tag{3}$$

By projecting centralized $(\hat{\boldsymbol{z}} - \boldsymbol{\mu})$ to the $\boldsymbol{U}_q$-subspace and re-projecting back, we can obtain the reconstructed features $\boldsymbol{U}_q \boldsymbol{U}_q^\top (\hat{\boldsymbol{z}} - \boldsymbol{\mu})$ and the reconstruction error $e(\hat{\boldsymbol{x}})$, which then can be set as the OoD detection score: $S(\hat{\boldsymbol{x}}) = -e(\hat{\boldsymbol{x}})$. An ideal case is that $\boldsymbol{U}_q$ contains informative principal components of InD data and causes projections of OoD data far away from that of InD data, leading to separable reconstruction errors between OoD and InD data.

## 3.2 Random Fourier features

A concise description is firstly given on the Random Fourier Features (RFFs) [13], which will be adopted in our method later. RFFs are proposed to approximate the kernel function so as to alleviate the heavy computation cost in large-scale kernel machines. In kernel methods, an $N \times N$ kernel matrix *w.r.t* $N$ samples requires $\mathcal{O}(N^2)$ kernel manipulations, $\mathcal{O}(N^2)$ space complexity and $\mathcal{O}(N^3)$ time complexity to calculate the inverse of the kernel matrix, which leads to overwhelmed computation costs for a large data size $N$. Therefore, RFFs are introduced by building an explicit feature mapping to directly approximate the kernel function for efficient kernel machines on large-scale data.

RFFs are built on the Bochner's theorem [36]: A continuous and shift-invariant kernel $k(\boldsymbol{z}_1, \boldsymbol{z}_2) = k(\boldsymbol{z}_1 - \boldsymbol{z}_2)$ on $\mathbb{R}^m$ is positive definite if and only if $k(\cdot)$ is the Fourier transform of a non-negative measure. An explicit feature mapping $\phi_{\mathrm{RFF}}$ induced from the kernel $k$ is derived in [13]:

$$\phi_{\mathrm{RFF}}(\boldsymbol{z}) \triangleq \sqrt{\frac{2}{M}} [\phi_1(\boldsymbol{z}), \cdots, \phi_M(\boldsymbol{z})], \quad \phi_i(\boldsymbol{z}) = \cos(\boldsymbol{z}^\top \boldsymbol{\omega}_i + u_i), i = 1, \cdots, M, \tag{4}$$

where $\boldsymbol{\omega}_1, \cdots, \boldsymbol{\omega}_M \in \mathbb{R}^m$ are *i.i.d.* sampled from the Fourier transform of $k(\cdot)$, and $u_1, \cdots, u_M \in \mathbb{R}$ are *i.i.d.* sampled from a uniform distribution $\mathcal{U}(0, 2\pi)$. For example, the sampling distribution for $\boldsymbol{\omega}_i$ of a Gaussian kernel function $k_{\mathrm{gau}} = e^{-\gamma \|\boldsymbol{z}_1 - \boldsymbol{z}_2\|_2^2}$ is $\boldsymbol{\omega} \sim \mathcal{N}(0, \sqrt{2\gamma}\boldsymbol{I}_m)$. Such a feature mapping $\phi_{\mathrm{RFF}}$ satisfies $k_{\mathrm{gau}}(\boldsymbol{z}_1, \boldsymbol{z}_2) \approx \phi_{\mathrm{RFF}}(\boldsymbol{z}_1)^\top \phi_{\mathrm{RFF}}(\boldsymbol{z}_2)$ and is known as the random Fourier features (RFFs) mapping. Refer to [13] for a detailed convergence analysis. RFFs have been widely utilized in kernel learning [37], optimization [38], etc.

# 4 Methodology

As empirically observed in [8], the aforementioned PCA reconstruction error in the $\boldsymbol{z}$-space is not an effective score in detecting OoD data from InD data. We propose that the reason behind is possibly due to the linearly-inseparable features of InD and OoD data, as shown in Figure 1a. To address this issue, we propose to explore the non-linearity in $\boldsymbol{z}$-space via kernel PCA. Then, through a kernel perspective on an existing KNN detector [7], we put forward two efficacious kernels that well characterize the non-linear patterns in $\boldsymbol{z}$-space of InD and OoD data: a cosine kernel (Section 4.1) and a cosine-Gaussian kernel (Section 4.2). Particularly, we adopt two explicit feature mappings $\Phi$ induced from the two kernels, and execute PCA in the *mapped* $\Phi(\boldsymbol{z})$-space, which leads to an informative principal subspace and distinct reconstruction errors for efficacious OoD detection.

## 4.1 Cosine kernel

In the KNN detector [7], the nearest neighbor searching is executed on the $\ell_2$-normalized penultimate features, *i.e.*, $\frac{\boldsymbol{z}}{\|\boldsymbol{z}\|_2}$. In inference, given a new sample $\hat{\boldsymbol{x}}$, its feature $\hat{\boldsymbol{z}}$ is firstly normalized as $\frac{\hat{\boldsymbol{z}}}{\|\hat{\boldsymbol{z}}\|_2}$,

then the negative of its ($k$-th) shortest $\ell_2$ distance to the $\ell_2$-normalized features $\frac{z_i}{\|z_i\|_2}$ of training data is set as the detection score:

$$S_{\text{knn}}(\hat{\boldsymbol{x}}) = - \min_{i:1,\cdots,N_{\text{tr}}} \left\| \frac{\hat{\boldsymbol{z}}}{\|\hat{\boldsymbol{z}}\|_2} - \frac{\boldsymbol{z}_i}{\|\boldsymbol{z}_i\|_2} \right\|_2. \tag{5}$$

The ablations in KNN demonstrate the indispensable significance of the $\ell_2$-normalization: the nearest neighbor searching directly on $\boldsymbol{z}$ shows a notably drop in detection performance. The critical role of the $\ell_2$-normalization in KNN attracts our attention in the sense of kernel. From a kernel perspective, the $\ell_2$-normalization is exactly the non-linear feature mapping $\phi_{\text{cos}}$ inducing the cosine kernel $k_{\text{cos}}$:

$$k_{\text{cos}}(\boldsymbol{z}_1, \boldsymbol{z}_2) = \frac{\boldsymbol{z}_1^\top \boldsymbol{z}_2}{\|\boldsymbol{z}_1\|_2 \cdot \|\boldsymbol{z}_2\|_2} = \phi_{\text{cos}}(\boldsymbol{z}_1)^\top \phi_{\text{cos}}(\boldsymbol{z}_2), \quad \phi_{\text{cos}}(\boldsymbol{z}) = \frac{\boldsymbol{z}}{\|\boldsymbol{z}\|_2}. \tag{6}$$

It indicates that a justified $\phi_{\text{cos}}(\boldsymbol{z})$-space with such non-linear mapping, instead of the original $\boldsymbol{z}$-space, contributes to the success of nearest neighbor searching in detecting OoD.

Notice that the key of KPCA for OoD detection lies in an appropriate non-linear feature space that captures the non-linearity in InD and OoD features, either through the kernel $k$ or the associated explicit feature mapping $\Phi$. Motivated by the KNN detector, we apply $\Phi(\cdot) \triangleq \phi_{\text{cos}}(\cdot)$ as the feature mapping in KPCA to introduce non-linearity. Then, PCA is executed on mapped features $\phi_{\text{cos}}(\boldsymbol{z})$, following the procedures described in Section 3.1. All the features $\boldsymbol{z}$ are now mapped to $\Phi(\boldsymbol{z})$ to formulate the covariance matrix $\boldsymbol{\Sigma}^\Phi$, for computing non-linear principal components with matrix $\boldsymbol{U}_q^\Phi$ and the corresponding reconstruction error $e^\Phi$. This detection scheme is dubbed as **CoP** (**Co**sine mapping followed by **P**CA), as shown in Algorithm 1. An in-depth analysis on the effect of the normalization of the cosine kernel is left in Appendix C.1.

## 4.2 Cosine-Gaussian kernel

The success of KNN (Equation (5)) suggests that the $\ell_2$ distance on $\frac{z}{\|z\|_2}$ is effective in distinguishing OoD data from InD data. In other words, the $\ell_2$ distance relation between samples in the $\phi_{\text{cos}}$-space preserves useful information that benefits the separation of OoD data from InD data. This motivates us to seek non-linear feature spaces that can retain the $\ell_2$ distance relation. Hence, we propose to introduce KPCA with non-linearity built upon $\phi_{\text{cos}}(\boldsymbol{z})$, through which the useful $\ell_2$ distance in $\phi_{\text{cos}}$-space can be preserved to further separate InD and OoD data.

In this regard, we deploy the shift-invariant Gaussian kernel to keep the $\ell_2$ distance information:

$$k_{\text{gau}}(\boldsymbol{z}_1, \boldsymbol{z}_2) = e^{-\gamma \|\boldsymbol{z}_1 - \boldsymbol{z}_2\|_2^2}. \tag{7}$$

The feature mapping associated with $k_{\text{gau}}$ is infinite-dimensional, but it can be efficiently approximated through random Fourier features (RFFs, [13]), *i.e.*, $\phi_{\text{RFF}}$ defined in Equation (4). In this way, the inner product of two mapped samples $\phi_{\text{RFF}}(\boldsymbol{z}_1)^\top \phi_{\text{RFF}}(\boldsymbol{z}_2)$ provides the approximate Gaussian kernel, so that we can leverage the RFFs mapping $\phi_{\text{RFF}}$ to preserve the $\ell_2$ distance information through the Gaussian kernel.

Hence, a cosine-Gaussian kernel is adopted for OoD detection, as the Gaussian kernel $k_{\text{gau}}$ (or $\phi_{\text{RFF}}$) is imposed on top of the cosine kernel $k_{\text{cos}}$ (or $\phi_{\text{cos}}$), further exploiting the $\ell_2$ distance relationships beyond the $\phi_{\text{cos}}$-space for OoD detection. As we work with the explicit feature mapping, the non-linearity to $\boldsymbol{z}$ is achieved by $\Phi(\cdot) \triangleq \phi_{\text{RFF}}(\phi_{\text{cos}}(\cdot))$. With mappings $\phi_{\text{RFF}}(\phi_{\text{cos}}(\boldsymbol{z}))$, PCA is then executed to compute the reconstruction errors for OoD detection. This detection scheme is dubbed as **CoRP** (**Co**sine and **R**FFs mappings followed by **P**CA). Algorithm 1 illustrates the complete procedure of the proposed **CoP** and **CoRP** for OoD detection. Alternative choices for more kernels are exploited in Appendix C.2.

To warp up, we devise two effective feature mappings induced from a cosine kernel and a cosine-Gaussian kernel to promote the separability of InD data and OoD data in non-linear feature spaces, inspired by effectiveness of the $\ell_2$ normalization and the $\ell_2$ distance from a kernel perspective on the KNN detector [7]. Our proposed two feature mappings well characterize the non-linearity in penultimate features $\boldsymbol{z}$ of DNNs between InD and OoD data, enabling PCA to extract an informative subspace *w.r.t* the mapped features through principal components and the reconstruction errors.

---

**Algorithm 1** Kernel PCA for Out-of-Distribution Detection

---

1: **if** CoP **then**
2:    $\Phi(\cdot) \leftarrow \phi_{\cos}(\cdot), \phi_{\cos}(\boldsymbol{z}) = \frac{\boldsymbol{z}}{\|\boldsymbol{z}\|_2}$.
3: **else if** CoRP **then**
4:    Sampling $\boldsymbol{\omega}_i \sim \mathcal{N}(0, \sqrt{2\gamma}\boldsymbol{I}_m), i = 1, \cdots, M$.
5:    Sampling $u_i \sim \mathcal{U}(0, 2\pi), i = 1, \cdots, M$.
6:    $\Phi(\cdot) \leftarrow \phi_{\mathrm{RFF}}(\phi_{\cos}(\cdot))$.
7: **end if**
8: Calculating the covariance matrix in the mapped $\Phi(\boldsymbol{z})$-space:
    $\boldsymbol{\Sigma}^\Phi = \sum_{i=1}^{N_{\mathrm{tr}}} (\Phi(\boldsymbol{z}_i) - \boldsymbol{\mu}^\Phi)(\Phi(\boldsymbol{z}_i) - \boldsymbol{\mu}^\Phi)^\top, \boldsymbol{\mu}^\Phi = \frac{1}{N_{\mathrm{tr}}} \sum_{i=1}^{N_{\mathrm{tr}}} \Phi(\boldsymbol{z}_i)$.
9: Applying eigendecomposition: $\boldsymbol{\Sigma}^\Phi = \boldsymbol{U}^\Phi \boldsymbol{\Lambda}^\Phi \boldsymbol{U}^{\Phi\top}$.
10: Taking the first $q$ columns of $\boldsymbol{U}^\Phi$ *w.r.t* the top-$q$ largest eigenvalues in $\boldsymbol{\Lambda}^\Phi$: $\boldsymbol{U}_q^\Phi = \boldsymbol{U}^\Phi[:,:q]$.
**Ensure:** Dimensionality-reduction matrix $\boldsymbol{U}_q^\Phi$.

11: Given a new sample $\hat{\boldsymbol{x}}$ and its features $\hat{\boldsymbol{z}}$.
12: Calculating the reconstruction error:
    $e^\Phi(\hat{\boldsymbol{x}}) = \left\| \boldsymbol{U}_q^\Phi \boldsymbol{U}_q^{\Phi\top} (\Phi(\hat{\boldsymbol{z}}) - \boldsymbol{\mu}^\Phi) - (\Phi(\hat{\boldsymbol{z}}) - \boldsymbol{\mu}^\Phi) \right\|_2$.
**Ensure:** Reconstruction error $e^\Phi$.

---

**Computation complexity**   In our method, given any new sample $\hat{\boldsymbol{x}}$ with the penultimate features $\hat{\boldsymbol{z}}$ in inference, to compute the reconstruction error $e^\Phi(\hat{\boldsymbol{x}})$, we only need the feature mapping $\Phi$, the projection matrix $\boldsymbol{U}_q^\Phi$ and the mean mapped training feature vector $\boldsymbol{\mu}^\Phi$. Both $\boldsymbol{U}_q^\Phi$ and $\boldsymbol{\mu}^\Phi$ can be pre-calculated and stored in preparation for inference. Therefore, the entire computation cost of CoP and CoRP comes from the construction of the explicit feature mapping $\Phi$ on new features $\hat{\boldsymbol{z}}$.

- For CoP, its feature mapping $\phi_{\cos}$ is an in-place operation and does not require additional computations. Therefore, the time and memory complexity of CoP is $\mathcal{O}(1)$.
- For CoRP, the feature mapping $\phi_{\mathrm{RFF}}$ of the Gaussian kernel requires $2M$ samplings for $\boldsymbol{\omega}_i$ and $u_i$, respectively, and $M$ dot-products and $M$ additions. Accordingly, the time and memory complexity of CoRP is $\mathcal{O}(M)$.

In contrast, regarding the Equation (5) of the KNN detector, all the training features have to be stored at hand and iterated in inference, which implies a heavy $\mathcal{O}(N_{\mathrm{tr}})$ time and memory complexity. The $\mathcal{O}(1)/\mathcal{O}(M)$ of CoP/CoRP significantly outperforms the $\mathcal{O}(N_{\mathrm{tr}})$ of KNN ($M \ll N_{\mathrm{tr}}$). Detailed empirical comparisons are provided in Section 5.1.

In the following, Section 5 exhibits the SOTA performance of our KPCA detector over multiple prevailing detection methods. Section 6 gives an analytical discussion between our covariance-based KPCA and classic KPCA via kernel functions for OoD detection. Due to space limitation, more in-depth investigations on the kernel properties are left in Appendix C, covering ablation studies (Appendix C.1), alternative kernels (Appendix C.2) and sensitivity analysis (Appendix C.3).

## 5   Experiments on OoD detection

In experiments, our KPCA-based detectors, CoP and CoRP, are firstly compared with KNN [7] in Section 5.1, and show stronger detection performance and cheaper computation costs. In Section 5.2, CoP and CoRP are further compared with the regularized PCA reconstruction error [8], and achieve SOTA OoD detection performance over various prevailing methods. The source code of this work has been publicly released[1]. All the experiments are executed on 1 NVIDIA GeForce RTX 3090 GPU.

**Datasets**   Experiments are executed on the commonly-used small-scale CIFAR10 [39] and large-scale ImageNet-1K benchmarks [40], following the settings in [7, 8]. For CIFAR10 as InD, OoD data sets include SVHN [41], LSUN [42], iSUN [43], Textures [44] and Places365 [45]. For ImageNet-1K as InD, OoD data sets contain iNaturalist [46], SUN [47], Places [45] and Textures [44].

---

[1]https://github.com/fanghenshaometeor/ood-kernel-pca

Table 1: The detection performance of different methods (**ResNet50** trained on **ImageNet-1K**).

| method | iNaturalist | | SUN | | Places | | Textures | | **AVERAGE** | |
|---|---|---|---|---|---|---|---|---|---|---|
| | FPR↓ | AUROC↑ | FPR↓ | AUROC↑ | FPR↓ | AUROC↑ | FPR↓ | AUROC↑ | FPR↓ | AUROC↑ |
| | | | | | **Standard Training** | | | | | |
| MSP [15] | 54.99 | 87.74 | 70.83 | 80.86 | 73.99 | 79.76 | 68.00 | 79.61 | 66.95 | 81.99 |
| ODIN [16] | 47.66 | 89.66 | 60.15 | 84.59 | 67.89 | 81.78 | 50.23 | 85.62 | 56.48 | 85.41 |
| Energy [5] | 55.72 | 89.95 | 59.26 | 85.89 | 64.92 | 82.86 | 53.72 | 85.99 | 58.41 | 86.17 |
| GODIN [27] | 61.91 | 85.40 | 60.83 | 85.60 | 63.70 | 83.81 | 77.85 | 73.27 | 66.07 | 82.02 |
| Mahala [23] | 97.00 | 52.65 | 98.50 | 42.41 | 98.40 | 41.79 | 55.80 | 85.01 | 87.43 | 55.47 |
| KNN [7] | 59.00 | 86.47 | 68.82 | 80.72 | 76.28 | 75.76 | 11.77 | 97.07 | 53.97 | 85.01 |
| CoP (ours) | 67.25 | 83.41 | 75.53 | 79.93 | 82.48 | 73.83 | 8.33 | 98.29 | 58.40 | 83.86 |
| CoRP (ours) | 50.07 | 89.32 | 62.56 | 83.74 | 72.76 | 78.91 | 9.02 | 98.14 | **48.60** | **87.53** |
| | | | | | **Supervised Contrastive Learning** | | | | | |
| MSP [15] | 32.18 | 93.30 | 60.36 | 84.21 | 61.68 | 83.94 | 50.62 | 84.68 | 51.21 | 86.53 |
| ODIN [16] | 23.48 | 95.80 | 50.73 | 88.43 | 53.99 | 87.30 | 41.88 | 88.60 | 42.52 | 90.03 |
| Energy [5] | 23.00 | 95.94 | 47.56 | 88.86 | 51.59 | 87.58 | 39.15 | 89.05 | 40.33 | 90.36 |
| SSD [33] | 57.16 | 87.77 | 78.23 | 73.10 | 81.19 | 70.97 | 36.37 | 88.52 | 63.24 | 80.09 |
| KNN [7] | 30.18 | 94.89 | 48.99 | 88.63 | 59.15 | 84.71 | 15.55 | 95.40 | 38.47 | 90.91 |
| CoP (ours) | 29.85 | 94.79 | 44.99 | 90.62 | 56.77 | 86.19 | 10.28 | 97.35 | **35.47** | **92.24** |
| CoRP (ours) | 23.61 | 95.86 | 41.07 | 91.25 | 53.52 | 87.27 | 10.23 | 97.04 | **32.11** | **92.86** |

Table 2: Comparisons on the computation complexity between KNN [7] and our CoRP (**ResNet50** on **ImageNet-1K**). Experiments are executed on the same machine for a fair comparison. The nearest neighbor searching of KNN is implemented via Faiss [48].

| method | time and memorty complexity | time consuming (ms, per sample) | storage |
|---|---|---|---|
| KNN | $\mathcal{O}(N_{\mathrm{tr}})$ | $\approx 15.59$ | $\approx 20$ GiB |
| CoP | $\mathcal{O}(1)$ | $\approx 0.035$ | $\approx 22$ MiB |
| CoRP | $\mathcal{O}(M)$ | $\approx 0.086$ | $\approx 29$ MiB |

**Metrics**   For the evaluation metrics, we employ the commonly-used (i) False Positive Rate of OoD samples with 95% true positive rate of InD samples (FPR), and (ii) Area Under the Receiver Operating Characteristic curve (AUROC). The *average* FPR95 and AUROC values over the selected multiple OoD data sets are viewed as the final comparison metrics.

## 5.1   Comparisons with nearest neighbor searching

The comparisons with KNN [7] cover both the benchmarks. Following the setups in KNN, for fair comparisons, we evaluate models trained via the standard cross-entropy loss and models trained via the supervised contrastive learning [49], and adopt the same checkpoints released by KNN: ResNet18 [50] on CIFAR10 and ResNet50 on ImageNet-1K. Here, the scoring function of CoP and CoRP is $S(\hat{\boldsymbol{x}}) = -e^{\Phi}(\hat{\boldsymbol{x}})$ in Algorithm 1.

Table 1 presents empirical results of ResNet50 on the ImageNet-1K benchmark. In the standard training, our CoRP shows superior detection performance over KNN with lower FPR and higher AUROC values averaged over multiple OoD data sets. In supervised contrastive learning, both CoP and CoRP outperform other baseline results on each OoD data set. These results show that the proposed KPCA exploring non-linear patterns is more advantageous than the nearest neighbor searching and all compared methods. Besides, the further improvements of CoRP over CoP also verify the effectiveness of the distance-preserving property of the Gaussian kernel $k_{\mathrm{gau}}$ on top of the cosine kernel $k_{\mathrm{cos}}$ for OoD detection.

On the other hand, regarding the computational complexity in inference, Table 2 empirically shows the superior $\mathcal{O}(1)/\mathcal{O}(M)$ time and memory complexity of CoP/CoRP over the $\mathcal{O}(N_{\mathrm{tr}})$ of KNN, including: (i) the inference time of the nearest neighbor search in KNN and the reconstruction error calculation in CoP/CoRP; (ii) the storage of the InD training features in KNN and the $\boldsymbol{U}_q^{\Phi}$ and $\boldsymbol{\mu}^{\Phi}$ in CoP/CoRP. To be specific, for KNN, storing and iterating all the $N_{\mathrm{tr}} = 1,281,167$ features of ImageNet-1K training set requires nearly 20 GiB and 16 ms, respectively, while our CoP and CoRP directly compute the reconstruction error for each new sample with the pre-calculated projection

Table 3: Comparisons on the detection performance between the regularized reconstruction error [8] and our CoP and CoRP fused with other OoD scores (MSP, Energy, ReAct and BATS) on each OoD data set (**ResNet50** trained on **ImageNet-1K**). Best average results are highlighted with underlines.

| method | OoD data sets | | | | | | | | **AVERAGE** | |
| | iNaturalist | | SUN | | Places | | Textures | | | |
| | FPR↓ | AUROC↑ | FPR↓ | AUROC↑ | FPR↓ | AUROC↑ | FPR↓ | AUROC↑ | FPR↓ | AUROC↑ |
|---|---|---|---|---|---|---|---|---|---|---|
| MSP [15] | 54.99 | 87.74 | 70.83 | 80.86 | 73.99 | 79.76 | 68.00 | 79.61 | 66.95 | 81.99 |
| + PCA [8] | 51.47 | 88.95 | 67.64 | 82.71 | 71.20 | 80.87 | 60.53 | 85.86 | 62.71 | 84.60 |
| **+ CoP** | **50.84** | **89.21** | **67.35** | **82.81** | **70.96** | **81.08** | **59.96** | **86.21** | **62.28** | **84.83** |
| **+ CoRP** | **43.70** | **91.70** | **61.79** | **85.43** | **66.67** | **83.07** | **45.67** | **91.86** | **54.46** | **88.02** |
| Energy [5] | 55.72 | 89.95 | 59.26 | 85.89 | 64.92 | 82.86 | 53.72 | 85.99 | 58.41 | 86.17 |
| + PCA [8] | 50.36 | 91.09 | 54.19 | 87.55 | 64.13 | 84.00 | 29.33 | 92.59 | 49.50 | 88.81 |
| **+ CoP** | **45.13** | **92.15** | **52.33** | **88.01** | **61.49** | **84.96** | **29.13** | **92.57** | **47.02** | **89.42** |
| **+ CoRP** | **26.85** | **95.15** | **40.38** | **90.76** | **51.26** | **87.35** | **12.11** | **97.17** | **32.65** | **92.61** |
| ReAct [18] | 20.38 | 96.22 | 24.20 | 94.20 | 33.85 | 91.58 | 47.30 | 89.80 | 31.43 | 92.95 |
| + PCA [8] | 10.17 | 97.97 | 18.50 | 95.80 | 27.31 | 93.39 | 18.67 | 95.95 | 18.66 | 95.76 |
| **+ CoP** | **13.30** | **97.44** | **19.80** | **95.37** | **29.92** | **92.64** | **15.90** | **96.51** | **19.73** | **95.49** |
| **+ CoRP** | **10.77** | **97.85** | **18.70** | **95.75** | **28.69** | **93.13** | **12.57** | **97.21** | **17.68** | **95.98** |
| BATS [19] | 42.26 | 92.75 | 44.70 | 90.22 | 55.85 | 86.48 | 33.24 | 93.33 | 44.01 | 90.69 |
| + PCA [8] | 29.66 | 94.49 | 38.11 | 90.03 | 51.70 | 87.25 | 13.46 | 97.09 | 33.23 | 92.56 |
| **+ CoP** | **27.14** | **94.87** | **34.36** | **91.96** | **47.68** | **87.87** | **11.97** | **97.33** | **30.29** | **93.01** |
| **+ CoRP** | **18.74** | **96.31** | **28.02** | **93.49** | **41.41** | **89.78** | **9.45** | **97.79** | **24.41** | **94.34** |
| ODIN [16] | 47.66 | 89.66 | 60.15 | 84.59 | 67.89 | 81.78 | 50.23 | 85.62 | 56.48 | 85.41 |
| Mahala [23] | 97.00 | 52.65 | 98.50 | 42.41 | 98.40 | 41.79 | 55.80 | 85.01 | 87.43 | 55.47 |
| ViM [35] | 68.86 | 87.13 | 79.62 | 81.67 | 83.81 | 77.80 | 14.95 | 96.74 | 61.81 | 85.83 |
| DICE [34] | 26.66 | 94.49 | 36.08 | 90.98 | 47.63 | 87.73 | 32.46 | 90.46 | 35.71 | 90.92 |
| DICE+ReAct | 20.08 | 96.11 | 26.50 | 93.83 | 38.34 | 90.61 | 29.36 | 92.65 | 28.57 | 93.30 |
| NNGuide [24] | 25.73 | 95.12 | 37.18 | 91.21 | 46.97 | 88.67 | 27.70 | 92.30 | 34.39 | 91.82 |
| DML+ [28] | 13.57 | 97.50 | 30.21 | 94.01 | 39.06 | 91.42 | 36.31 | 89.70 | 29.79 | 93.16 |
| ASH-B [20] | 14.21 | 97.32 | 22.08 | 95.10 | 33.45 | 92.31 | 21.17 | 95.50 | 22.73 | 95.06 |
| ASH-S [20] | 11.49 | 97.87 | 27.98 | 94.02 | 39.78 | 90.98 | 11.93 | 97.60 | 22.80 | 95.12 |
| SCALE [21] | 9.50 | 98.17 | 23.27 | 95.02 | 34.51 | 92.26 | 12.93 | 97.37 | 20.05 | 95.71 |
| DDCS [22] | 11.63 | 97.85 | 18.63 | 95.68 | 28.78 | 92.89 | 18.40 | 95.77 | 19.36 | 95.55 |

matrix and the mean vector from the training data, resulting in a much higher processing speed and far less storage. The number of RFFs $M$ for CoRP in this experiment is $M = 4,096$ ($M \ll N_{\mathrm{tr}}$).

In addition, KPCA also outperforms KNN on the CIFAR10 benchmark with improved OoD detection performances, which we leave to Appendix B for more details.

## 5.2 Comparisons with regularized reconstruction errors

In [8], to alleviate the weak detection performance of PCA reconstruction error $e(\hat{z})$ of Equation (3), the authors proposed to *regularize* $e(\hat{z})$ by the feature norm $\|\hat{z}\|_2$ and a fusion strategy to boost its detection performance by introducing existing OoD scores. Firstly, the regularized reconstruction error $e_{\mathrm{reg}}(\hat{z})$ is calculated in the original $z$-space as : $e_{\mathrm{reg}}(\hat{z}) = \frac{\|U_q U_q^\top (\hat{z}-\mu)-(\hat{z}-\mu)\|_2}{\|\hat{z}\|_2}$. Then, the authors claimed that such a regularized version $e_{\mathrm{reg}}(\hat{z})$ is still insufficient for OoD detection, and designed a fusion strategy to combine $e_{\mathrm{reg}}$ with other existing OoD scores. For example, to fuse $e_{\mathrm{reg}}$ with the Energy [5] score, the final scoring function is $S(\hat{x}) = (1 - e_{\mathrm{reg}}(\hat{z})) \cdot S_{\mathrm{Energy}}(\hat{z})$.

In this section, we show that our KPCA reconstruction error $e^\Phi$ outperforms the regularized PCA reconstruction error $e_{\mathrm{reg}}$ under the same fusion framework. Following the settings in [8], for a fair comparison, the fused OoD scores include MSP [15], Energy [5], ReAct [18] and BATS [19]. The detection experiments are executed on the ImageNet-1K benchmark with pre-trained ResNet50 and MobileNet [51] checkpoints from PyTorch [52].

Table 3 presents the comparisons between [8] and ours on the ImageNet-1K benchmark of ResNet50. When fused with MSP, Energy and BATS, both the KPCA-based CoP and CoRP outperform the regularized reconstruction error [8] on almost all the OoD data sets with substantially improved FPR and AUROC values. Specifically, when fused with the ReAct method [18], the CoRP achieves new SOTA OoD detection performance among various prevailing detectors. Experiments on MobileNet also show superior performance of CoP and CoRP, see details in Appendix B.

All these experiment results indicate that an appropriately mapped $\Phi(\boldsymbol{z})$-space benefits the OoD detection, as the non-linearity in $\boldsymbol{z}$-space gets alleviated by the feature mapping $\Phi$. Our work provides 2 viable selections for $\Phi$ with empirical validations, which we hope could attract attentions towards the non-linearity in InD and OoD features for the research community from a kernel perspective.

## 6 Analytical discussions with KPCA via kernel functions

In CoP and CoRP, KPCA is executed with the *covariance matrix* of *mapped* features $\Phi(\boldsymbol{z})$. In contrast, in the classic KPCA [11, 12], such feature mappings $\Phi$ are not explicitly given, and it rather works with a *kernel function* applied to *original* features $\boldsymbol{z}$. In this section, we supplement our covariance-based KPCA with its kernel function implementation, including theoretical discussions and empirical comparisons on OoD detection. Our CoP and CoRP are shown to be more effective and efficient than their counterparts that employ kernel functions.

In the classic KPCA, the kernel trick enables projections to the principal subspace via kernel functions without calculating $\Phi$. However, how to map the projections in the principal subspace back to the original $\boldsymbol{z}$-space remains a non-trivial issue, known as the pre-image problem [53], which makes it problematic to calculate reconstructed features via kernel functions. To address this issue, the following Proposition 1 shows a flexible way to directly calculate reconstruction errors without building reconstructed features, so as to apply the kernel trick, shown in the subsequent Proposition 2.

**Proposition 1.** *The KPCA reconstruction error $e^{\Phi}(\hat{\boldsymbol{z}})$ can be represented as the norm of features projected in the residual subspace, i.e., the $\boldsymbol{U}_p^{\Phi}$-subspace with $\boldsymbol{U}^{\Phi} = [\boldsymbol{U}_q^{\Phi}, \boldsymbol{U}_p^{\Phi}]$:*

$$e^{\Phi}(\hat{\boldsymbol{z}}) = \|\boldsymbol{U}_p^{\Phi\top}(\Phi(\hat{\boldsymbol{z}}) - \boldsymbol{\mu}^{\Phi})\|_2. \tag{8}$$

Proposition 1 implies that the reconstruction error equals to the norm of projections in the residual $\boldsymbol{U}_p^{\Phi}$-subspace, *i.e.*, the subspace consisting of those principal components that are not kept, see proofs in Appendix D. Accordingly, as typically done in the classic KPCA, we can introduce a kernel function to perform dimension reduction, but to the residual subspace, and then calculate the norms of the reduced features as the reconstruction error, illustrated by Proposition 2.

Given a kernel function $k(\cdot, \cdot) : \mathbb{R}^m \times \mathbb{R}^m \to \mathbb{R}$, we have a kernel matrix $\boldsymbol{K} \in \mathbb{R}^{N_{\mathrm{tr}} \times N_{\mathrm{tr}}}$ on training data with $\boldsymbol{K}_{i,j} = k(\boldsymbol{z}_i, \boldsymbol{z}_j)$, and a vector $\boldsymbol{k}_{\hat{\boldsymbol{z}}} \in \mathbb{R}^{N_{\mathrm{tr}}}$ with the $i$-th element $k(\boldsymbol{z}_i, \hat{\boldsymbol{z}})$ for a new sample $\hat{\boldsymbol{z}}$. Proposition 2 shows how to calculate the KPCA reconstruction error via the kernel function $k$.

**Proposition 2.** *The KPCA reconstruction error $e^k(\hat{\boldsymbol{z}})$ w.r.t a kernel function $k$ can be calculated as:*

$$e^k(\hat{\boldsymbol{z}}) = \|\boldsymbol{A}^{\top}\boldsymbol{k}_{\hat{\boldsymbol{z}}}\|_2, \tag{9}$$

*where $\boldsymbol{A} \in \mathbb{R}^{N_{\mathrm{tr}} \times l}$ includes $l$ eigenvectors of the kernel matrix $\boldsymbol{K}$ w.r.t the top-$l$ smallest eigenvalues.*

According to Proposition 2, now CoP and CoRP can be implemented via kernel functions. For CoP, we just directly apply the cosine kernel function $k_{\cos}$ on features $\boldsymbol{z}$ to compute the kernel matrix $\boldsymbol{K}$ and the projection matrix $\boldsymbol{A}$, so as to obtain $e^k(\hat{\boldsymbol{z}})$ following Equation (9). For CoRP, we should adopt the Gaussian kernel function $k_{\mathrm{gau}}$ on the $\ell_2$-normalized inputs $\frac{\boldsymbol{z}}{\|\boldsymbol{z}\|_2}$ to calculate $\boldsymbol{K}$, $\boldsymbol{A}$ and $e^k(\hat{\boldsymbol{z}})$. Figure 2 shows comparisons on the detection performance between CoP/CoRP and their kernel function implementations.

In Figure 2, the detection performance of KPCA with kernel functions is evaluated by varying the explained variance ratio of the kernel matrix $\boldsymbol{K}$. The larger the explained variance ratio, the smaller the dimension $l$ of $\boldsymbol{A}$. The best detection results achieved by CoP/CoRP are illustrated as the dashed lines. Clearly, regarding the OoD detection performance, reconstruction errors $e^k$ calculated by kernel functions are less effective than those calculated explicitly in the mapped $\Phi(\boldsymbol{z})$-space.

Aside from the detection performance, KPCA with kernel functions is far less computationally efficient than CoP/CoRP in two aspects. On the one hand, the time expense of eigendecomposition on the $N_{\mathrm{tr}} \times N_{\mathrm{tr}}$ kernel matrix $\boldsymbol{K}$ by the former is more expensive than that on the $m \times m$ or $M \times M$ covariance matrix $\boldsymbol{\Sigma}^{\Phi}$ by the latter, since $N_{\mathrm{tr}} \gg M$ and $N_{\mathrm{tr}} \gg m$. For example, on the ImageNet-1K benchmark with MobileNet, these settings are $N_{\mathrm{tr}} = 1,281,167$, $M = 2560$ and $m = 1280$, on which KPCA with kernel functions is actually nearly prohibitive. On the other hand, in the inference stage, KPCA via kernel functions yet requires an $\mathcal{O}(N_{\mathrm{tr}})$ time and memory complexity in calculating $\boldsymbol{k}_{\hat{\boldsymbol{z}}}$, as all the training data has to be stored and iterated, which is much higher than the $\mathcal{O}(1)/\mathcal{O}(M)$ complexity of our CoP/CoRP.

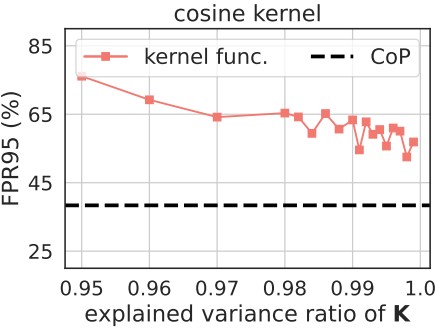 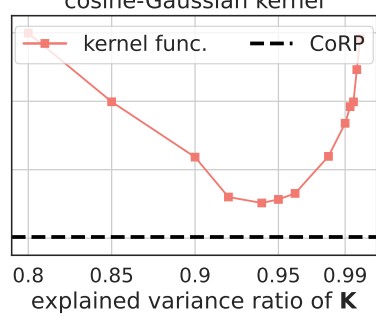

Figure 2: Comparisons on the average detection FPR values between CoP/CoRP and their kernel function implementations in the CIFAR10 benchmark. In experiments, 5,000 images of the CIFAR10 training set and 1,000 images of the CIFAR10 test set and OoD data sets are randomly selected.

# 7 Conclusion

As PCA reconstruction errors fail to distinguish OoD data from InD data on the penultimate features $z$ of DNNs, kernel PCA is introduced for its non-linearity in the manner of employing explicit feature mappings. To find appropriate kernels that can characterize the non-linear patterns in InD and OoD features, we take a kernel perspective to decouple and analyze key components of an existing KNN detector [7], and thus propose a cosine kernel and a cosine-Gaussian kernel for KPCA. Specifically, two explicit feature mappings $\Phi(\cdot)$ induced from the two kernels are leveraged on original features $z$. For the cosine kernel, its explicit feature mapping can be directly obtained. For the Gaussian kernel, we adopt the celebrated random Fourier features to approximate the Gaussian kernel. The mapped $\Phi(z)$-space enables PCA to extract principal components that well separate InD and OoD data, leading to distinguishable reconstruction errors. Extensive empirical results have verified the improved effectiveness and efficiency of the proposed KPCA with new SOTA OoD detection performance. Besides, more in-depth analyzes are drawn on the individual effects of the cosine kernel and the Gaussian kernel, and the involved multiple hyper-parameters. In addition, theoretical discussions and associated experiments are provided to bridge the relationships between our covariance-based KPCA and its kernel function implementation so as to further illustrate the advantages of our method.

One limitation of the KPCA detector is that the two specific kernels are still manually selected with carefully-tuned parameters. It remains a valuable topic in the OoD detection task whether the parameters of kernels could be learned from data according to some optimization objective. For example, deep kernel learning [54] could be considered as an alternative choice so as to pursue stronger kernels that can better characterize InD and OoD with enhanced detection performance by an additional learning step on the features. We hope that the proposed two effective kernels verified empirically in our work could benefit the research community as a solid example for future studies.

## Acknowledgments and Disclosure of Funding

This work is jointly supported by National Natural Science Foundation of China (62376153, 62376155), and Shanghai Municipal Science and Technology Research Major Project (2021SHZDZX0102).

## Societal impacts

The societal impacts of this work are mainly positive, as it aims at detecting OoD samples in the inference or deployment stage of DNNs, which benefits researches in trustworthy deep learning. Through our work, we hope that new inspirations on the non-linearity in data could be drawn from a kernel perspective so as to highlight the safety issue in real-world machine learning applications.

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

# A   Details of representative OoD detectors

In this section, we elaborate the scoring function $S(\cdot)$ of the OoD detectors included in the comparison experiments of Section 5. Given a well-trained DNN $f : \mathbb{R}^d \to \mathbb{R}^c$ with inputs $x \in \mathbb{R}^d$, the outputs are $c$-dimension logits $f(x) \in \mathbb{R}^c$ *w.r.t* $c$ classes. The DNN $f$ learns features $z \in \mathbb{R}^m$ of $x$ before the last linear layer, *i.e.*, the penultimate features $z$.

**MSP** [15] employs the softmax function on the output logits and takes the maximum probability as the scoring function. Given a new sample $\hat{x} \in \mathbb{R}^d$, its MSP score is

$$S_{\text{MSP}}(\hat{x}) = \max\left(\text{softmax}(f(\hat{x}))\right). \tag{10}$$

**ODIN** [16] introduces the temperature scaling and adversarial examples into the MSP score:

$$S_{\text{ODIN}}(\hat{x}) = \max\left(\text{softmax}(\frac{f(\hat{x}_{\text{a}})}{T})\right), \tag{11}$$

where $T$ denotes the temperature and $\hat{x}_{\text{a}}$ denotes the perturbed adversarial examples of $\hat{x}$.

**Mahala** [23] employs the Mahalanobis score to perform OoD detection. The DNN outputs at different layers are modeled as a mixture of multivariate Gaussian distributions, and the Mahalanobis distance is calculated. Then, a linear regressor is trained to achieve a weighted Mahalanobis distance at different layers as the final detection score. To train the linear regressor, the training data and the corresponding adversarial examples are adopted as positive and negative samples, respectively.

$$S_{\text{mahal}}^l(\hat{x}) = \max_i \; -(f^l(\hat{x}) - \mu_i^l)^T \Sigma_l (f^l(\hat{x}) - \mu_i^l),$$
$$S_{\text{mahal}}(\hat{x}) = \sum_l \alpha_l \cdot S_{\text{mahal}}^l(\hat{x}), \tag{12}$$

where $f^l(\hat{x})$ denotes the output features at the $l$-th layer with the associated mean feature vector $\mu_i^l$ of class-$i$ and the covariance matrix $\Sigma_l$, and $\alpha_l$ denotes the linear regression coefficients.

**Energy** [5] uses an energy function on logits as energy is well aligned with input probability densities:

$$S_{\text{energy}}(\hat{x}) = \log \sum_{i=1}^{c} \exp(f_i(\hat{x})), \tag{13}$$

where $f_i(\hat{x})$ denotes the $i$-th element in the $c$-dimension output logits $f(\hat{x})$.

**GODIN** [27] improves ODIN from 2 aspects: decomposing the probabilities and modifying the input pre-processing. On the one hand, a two-branch structure with learnable parameters is imposed after the logits to formulate the decomposed probablities. On the other hand, the magnitude of the adversarial examples is optimized instead of manually tuned in ODIN.

**ReAct** [18] proposes activation truncation on the penultimate features $z$ of DNNs, as the authors observe that features of OoD data generally hold high unit activations in the penultimate layers. The feature clipping is implemented in a simple way:

$$\bar{z} = \min\{z, \alpha\}, \tag{14}$$

where $\alpha$ is a pre-defined constant. The clipped features $\bar{z}$ then pass through the last linear layer and yield modified logits. Other logits-based OoD methods such as Energy could be applied on the modified logits to produce a detection score.

**KNN** [7] is a simple but time-consuming and memory-inefficient detector since it performs nearest neighbor search on the $\ell_2$-normalized penultimate features between the test sample and all the training samples. The negative of the ($k$-th) shortest $\ell_2$ distance is set as the score for a new sample $\hat{x}$.

**ViM** [35] combines information from both logits and features in a complicated way for OoD detection. Firstly, penultimate features $z$ are projected to the residual space obtained by PCA. Then the norm of projected features gets scaled together with the logits via the softmax function. Finally the scaled feature norm is selected as the detection score.

**DICE** [34] is a sparsification-based OoD detector by preserving the most important weights in the last linear layer. Denote the weights $W \in \mathbb{R}^{m \times c}$ and the bias $b \in \mathbb{R}^c$ in the last linear layer, the forward propagation of DICE is defined as:

$$f_{\text{DICE}}(\hat{x}) \triangleq (M \odot W)^\top \hat{z} + b. \tag{15}$$

$\odot$ is the element-wise multiplication, and $M \in \mathbb{R}^{m \times c}$ is a masking matrix whose elements are determined by the element-wise multiplication between the $i$-th column $w_i \in \mathbb{R}^m$ in $W$ and the penultimate features $\hat{z}$: $w_i \odot \hat{z}$. Then, similar as ReAct, logits-based detectors could be executed on the modified logits $f_{\text{DICE}}(\hat{x})$ to produce a detection score.

**BATS** [19] proposes to truncate the extreme outputs of Batch Normalization (BN) layers via the estimated mean and standard deviations stored in BN layers, as those extreme features would lead to ambiguity and should be rectified. However, in the released code, the authors actually does not use any information from the BN layers, but instead simply perform clipping on the penultimate features $z$ via the feature mean and standard deviations.

**PCA** [8] re-formulates the reconstruction errors and empirically shows the inseparativity via the re-formulated errors between InD and OoD data in the primal $z$-space. The authors further propose a regularized reconstruction error and a fusion strategy to boost the OoD detection performance.

**NNGuide** [24] exploits the guidance of the Energy score on logits to boost the detection performance of the nearest neighbor search on features. Specifically, the training features are firstly scaled by their corresponding Energy scores, then KNN is executed on such re-scaled features for the new sample. The final detection score is set as the multiplication of the searched distance and the Energy score. The time and memory complexity of NNGuide is still the same $\mathcal{O}(N_{\text{tr}})$ as that of KNN [7].

**DML** [28] decouples the maximum logits into two parts: the maximum cosine similarity (MaxCosine) and the maximum norm (MaxNorm), and employs their ensemble as the detection score. DML reveals that the cosine similarity and the feature norm jointly contribute to the effectiveness of the previous MSP [15] and MaxLogit [14] methods and designs new training losses from the perspective of feature collapse [55], so as to further improve the performance of MaxCosine and MaxNorm.

**ASH** [20] is a post-hoc detection method that removes the abnormal information in features. ASH includes two stages: removing a large portion of the features, and adjusting the remaining feature values by scaling up or assigning a constant value. ASH exhibits advantages over the classic ReAct method [18]: no global thresholds and stronger flexibility, and shows better detection performance.

**SCALE** [21] analyzes the rectification and scaling components of the ASH method, and improves ASH by only a scaling process in the post-hoc stage.

**DDCS** [22] investigates the effects of different channels based on existing feature-clipping detection methods, and proposes a channel-level anomalous activations pre-rectifying module so as to clip features more carefully for better detection performance.

We follow the settings in KNN [7] and include **CSI** [32] and **SSD** [33] into the comparisons in Section 5.1. The 2 methods adopt the contrastive losses to train DNNs. In the comparison results of Table 1 and the following Table 4, the reported detection results of CSI and SSD are directly from [7], and are obtained by executing the Mahalanobis detector on learned features of DNNs trained by CSI and SSD. Refer to [7] for more details.

# B    Supplementary experiment results on OoD detection

Table 4 illustrates the comparison results on the CIFAR10 benchmark between our CoP/CoRP and the KNN detector [7]. Similar as the comparisons on the ImageNet-1K benchmark in Table 1, CoRP outperforms other baselines in both the standard training and the supervised contrastive learning with lower FPR and higher AUROC average values.

Table 5 shows the comparison results on the ImageNet-1K benchmark between our CoP/CoRP and the regularized reconstruction error [8] of MobileNet [51]. Similar as the case of ResNet50 in Table 3, under the same fusion trick with other detection scores, our KPCA reconstruction errors of CoP and CoRP significantly outperform the regularized PCA reconstruction error of [8], implying the substantial improvements by characterizing the non-linear data distribution of the InD and OoD features via the devised two proper non-linear kernels.

Table 4: The detection performance of different methods (**ResNet18** trained on **CIFAR10**).

| method | SVHN | | LSUN | | iSUN | | Textures | | Places365 | | AVERAGE | |
|---|---|---|---|---|---|---|---|---|---|---|---|---|
| | FPR↓ | AUROC↑ | FPR↓ | AUROC↑ | FPR↓ | AUROC↑ | FPR↓ | AUROC↑ | FPR↓ | AUROC↑ | FPR↓ | AUROC↑ |
| | | | | | **Standard Training** | | | | | | | |
| MSP [15] | 59.66 | 91.25 | 45.21 | 93.80 | 54.57 | 92.12 | 66.45 | 88.50 | 62.46 | 88.64 | 57.67 | 90.86 |
| ODIN [16] | 20.93 | 95.55 | 7.26 | 98.53 | 33.17 | 94.65 | 56.40 | 86.21 | 63.04 | 86.57 | 36.16 | 92.30 |
| Energy [5] | 54.41 | 91.22 | 10.19 | 98.05 | 27.52 | 95.59 | 55.23 | 89.37 | 42.77 | 91.02 | 38.02 | 93.05 |
| GODIN [27] | 15.51 | 96.60 | 4.90 | 99.07 | 34.03 | 94.94 | 46.91 | 89.69 | 62.63 | 87.31 | 32.80 | 93.52 |
| Mahala [23] | 9.24 | 97.80 | 67.73 | 73.61 | 6.02 | 98.63 | 23.21 | 92.91 | 83.50 | 69.56 | 37.94 | 86.50 |
| KNN [7] | 24.53 | 95.96 | 25.29 | 95.69 | 25.55 | 95.26 | 27.57 | 94.71 | 50.90 | 89.14 | 30.77 | 94.15 |
| CoP (ours) | 11.56 | 97.57 | 23.24 | 95.56 | 53.71 | 88.74 | 26.28 | 93.87 | 74.11 | 80.24 | 37.78 | 91.20 |
| CoRP (ours) | 20.68 | 96.53 | 19.19 | 96.71 | 21.49 | 96.26 | 21.61 | 96.08 | 53.73 | 89.14 | **27.34** | **94.95** |
| | | | | | **Supervised Contrastive Learning** | | | | | | | |
| CSI [32] | 37.38 | 94.69 | 5.88 | 98.86 | 10.36 | 98.01 | 28.85 | 94.87 | 38.31 | 93.04 | 24.16 | 95.89 |
| SSD [33] | 1.51 | 99.68 | 6.09 | 98.48 | 33.60 | 95.16 | 12.98 | 97.70 | 28.41 | 94.72 | 16.52 | 97.15 |
| KNN [7] | 2.42 | 99.52 | 1.78 | 99.48 | 20.06 | 96.74 | 8.09 | 98.56 | 23.02 | 95.36 | 11.07 | 97.93 |
| CoP (ours) | 0.55 | 99.85 | 1.12 | 99.67 | 23.91 | 96.11 | 4.79 | 99.06 | 19.92 | 95.63 | **10.06** | **98.07** |
| CoRP (ours) | 0.74 | 99.82 | 0.89 | 99.75 | 13.08 | 97.36 | 4.59 | 99.03 | 17.44 | 95.89 | **7.35** | **98.37** |

Table 5: Comparisons on the detection performance between the regularized reconstruction error [8] and our CoP and CoRP fused with other OoD scores (MSP, Energy, ReAct and BATS) on each OoD data set (**MobileNet** trained on **ImageNet-1K**). Best average results are highlighted with underlines.

| method | iNaturalist | | SUN | | Places | | Textures | | AVERAGE | |
|---|---|---|---|---|---|---|---|---|---|---|
| | FPR↓ | AUROC↑ | FPR↓ | AUROC↑ | FPR↓ | AUROC↑ | FPR↓ | AUROC↑ | FPR↓ | AUROC↑ |
| MSP [15] | 64.29 | 85.32 | 77.02 | 77.10 | 79.23 | 76.27 | 73.51 | 77.30 | 73.51 | 79.00 |
| + PCA [8] | 59.49 | 86.87 | 73.75 | 79.41 | 76.79 | 77.94 | 65.71 | 83.46 | 68.93 | 81.92 |
| **+ CoP** | **57.14** | **87.62** | **72.86** | **79.45** | **76.17** | **77.77** | **60.71** | **86.42** | **66.72** | **82.82** |
| **+ CoRP** | **55.71** | **88.10** | **71.48** | **80.77** | **75.33** | **78.90** | **58.90** | **87.13** | **65.36** | **83.73** |
| Energy [5] | 59.50 | 88.91 | 62.65 | 84.50 | 69.37 | 81.19 | 58.05 | 85.03 | 62.39 | 84.91 |
| + PCA [8] | 56.92 | 89.62 | 60.07 | 85.80 | 69.23 | 81.72 | 34.22 | 91.66 | 55.11 | 87.20 |
| **+ CoP** | **51.21** | **90.79** | **59.88** | **85.84** | **68.62** | **81.74** | **23.16** | **94.55** | **50.72** | **88.23** |
| **+ CoRP** | **43.85** | **91.96** | **52.17** | **87.91** | **63.75** | **83.59** | **19.02** | **95.41** | **44.70** | **89.72** |
| ReAct [18] | 43.07 | 92.72 | 52.47 | 87.26 | 59.91 | 84.07 | 40.20 | 90.96 | 48.91 | 88.75 |
| + PCA [8] | 35.84 | 93.66 | 40.35 | 90.77 | 52.38 | 86.76 | 18.44 | 95.39 | 36.75 | 91.65 |
| **+ CoP** | **35.84** | **93.54** | 48.12 | 88.97 | 60.62 | 84.45 | **12.62** | **96.97** | 39.30 | 90.98 |
| **+ CoRP** | **31.72** | **94.27** | 40.77 | **90.98** | 55.69 | 86.42 | **10.48** | **97.49** | **34.66** | **92.29** |
| BATS [19] | 49.57 | 91.50 | 57.81 | 85.96 | 64.48 | 82.83 | 39.77 | 91.17 | 52.91 | 87.87 |
| + PCA [8] | 50.51 | 90.86 | 55.41 | 87.00 | 66.43 | 82.60 | 23.26 | 94.70 | 48.90 | 88.79 |
| **+ CoP** | **42.68** | **92.24** | **55.01** | 86.89 | **65.70** | 82.44 | **13.78** | **96.77** | **44.29** | **89.58** |
| **+ CoRP** | **36.10** | **93.37** | **45.92** | **89.47** | **59.82** | **84.83** | **11.37** | **97.24** | **38.30** | **91.23** |
| ODIN [16] | 58.54 | 87.51 | 57.00 | 85.83 | 59.87 | 84.77 | 52.07 | 85.04 | 56.87 | 85.79 |
| Mahala [23] | 62.11 | 81.00 | 47.82 | 83.66 | 52.09 | 83.63 | 92.38 | 33.06 | 63.60 | 71.01 |
| ViM [35] | 91.83 | 77.47 | 94.34 | 70.24 | 93.97 | 68.26 | 37.62 | 92.65 | 79.44 | 77.15 |
| DICE [34] | 43.28 | 90.79 | 38.86 | 90.41 | 53.48 | 85.67 | 33.14 | 91.26 | 42.19 | 89.53 |
| DICE+ReAct | 41.75 | 89.84 | 39.07 | 90.39 | 54.41 | 84.03 | 19.98 | 95.86 | 38.80 | 90.03 |
| NNGuide [24] | 45.73 | 91.19 | 51.03 | 87.87 | 60.60 | 84.44 | 29.50 | 92.47 | 46.72 | 88.99 |
| ASH-B [20] | 31.46 | 94.28 | 38.45 | 91.61 | 51.80 | 87.56 | 20.92 | 95.07 | 35.66 | 92.13 |
| ASH-S [20] | 39.10 | 91.94 | 43.62 | 90.02 | 58.84 | 84.73 | 13.12 | 97.10 | 38.67 | 90.95 |

# C Analytical discussions on kernels

In this section, more in-depth discussions are drawn on the effects of cosine and Gaussian kernels in CoP and CoRP for OoD detection in Appendix C.1 and Appendix C.2, respectively. A comprehensive sensitivity analysis on the involved hyper-parameters in CoP and CoRP are presented in Appendix C.3.

## C.1 Effects of the cosine kernel

The cosine kernel in CoP and CoRP appears an indispensable basis in alleviating the linear inseparability in InD and OoD features. The reason for its effectiveness lies in the imbalanced feature norms $\|z\|_2$ between InD and OoD features, which has also been observed in preceding works [7, 6, 32]. Figure 3 shows the feature norms of multiple InD and OoD data sets, from which one can find clear disparities of the InD and OoD feature norms. The cosine kernel $k_{\cos}$ and the corresponding feature

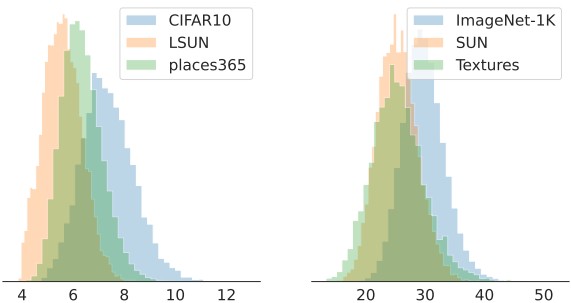

Figure 3: A density histogram of the imbalanced norms of InD and OoD features. InD: CIFAR10 and ImageNet-1K. OoD: LSUN and places365, SUN and Textures.

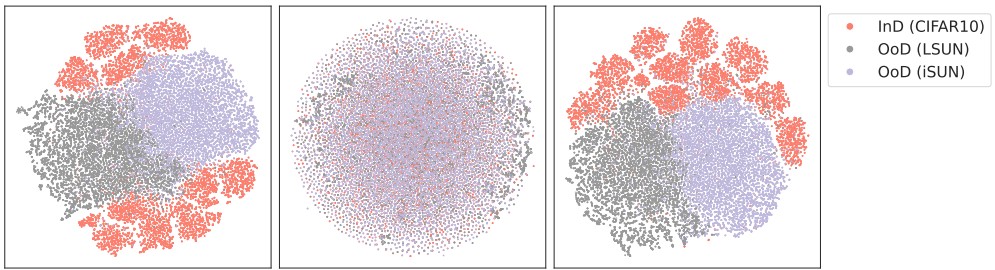

Figure 4: T-SNE visualization of the original features (left), mapped features *w.r.t* a Gaussian kernel (middle) and mapped features *w.r.t* a cosine kernel (right).

mapping $\phi_{\cos}$ in Equation (6) thereby normalize the feature norms and facilitate the separability between InD and OoD data.

Figure 4 illustrates the t-SNE visualization on the InD and OoD features to further imply the importance of the cosine kernel. As shown in the middle panel of Figure 4, the Gaussian kernel alone fails in creating separable InD and OoD features in the mapped space and actually leads to a complete mess of the mapped features. In contrast, the cosine kernel significantly alleviates the linearly-inseparability of features, shown in the right panel of Figure 4.

Ablation studies on the cosine feature mapping $\phi_{\cos}$ in CoP and CoRP are executed to verify its indispensability for OoD detection. Specifically, CoP without $\phi_{\cos}$ reduces to a standard PCA on the $\boldsymbol{z}$-space, and CoRP without $\phi_{\cos}$ reduces to KPCA with a Gaussian kernel. Table 6 shows the corresponding detection FPR and AUROC values on each OoD data set in the ImageNet-1K benchmark of ablations on $\phi_{\cos}$. Both the standard PCA and the Gaussian KPCA exhibit worse detection performance than CoP and CoRP. Particularly, the single Gaussian kernel in KPCA actually results in a complete failure in detecting OoD samples with nearly 95% FPR values. Therefore, the cosine kernel is essential in characterizing the non-linearity in InD and OoD features and critical for the superior performance of the KPCA detector.

## C.2 Alternative kernels

The design of the Gaussian kernel in CoRP is motivated by the useful $\ell_2$ distance on $\frac{\boldsymbol{z}}{\|\boldsymbol{z}\|_2}$ in the KNN detector [7]. The Gaussian kernel preserves the beneficial $\ell_2$ distance relationship in $\phi_{\cos}(\boldsymbol{z})$-space through the RFFs mapping. In this section, we provide two alternative choices aside from the cosine-Gaussian kernel.

- The cosine-Laplacian kernel explores the $\ell_1$ distance in $\phi_{\cos}(\boldsymbol{z})$-space via the Laplacian kernel $k_{\text{lap}}$:

$$k_{\text{lap}}(\boldsymbol{z}_1, \boldsymbol{z}_2) = e^{-\gamma\|\boldsymbol{z}_1 - \boldsymbol{z}_2\|_1}. \tag{16}$$

To construct the RFFs for $k_{\text{lap}}$, the sampling distribution of the $\omega_i$ in Equation (4) is a Cauchy distribution $\omega \sim p(\omega) = \frac{\gamma^2}{\pi\gamma(\omega^2 + \gamma^2)}$.

Table 6: The detection results among a variety of kernels (**ResNet50** trained on **ImageNet-1K**).

| kernel | OoD data sets | | | | | | | | AVERAGE | |
| | iNaturalist | | SUN | | Places | | Textures | | | |
| | FPR↓ | AUROC↑ | FPR↓ | AUROC↑ | FPR↓ | AUROC↑ | FPR↓ | AUROC↑ | FPR↓ | AUROC↑ |
|---|---|---|---|---|---|---|---|---|---|---|
| PCA (no kernels) | 95.46 | 52.01 | 97.98 | 44.86 | 97.99 | 45.19 | 46.22 | 87.77 | 84.41 | 57.46 |
| Polynomial | 96.03 | 53.07 | 98.26 | 42.84 | 97.85 | 45.02 | 95.50 | 47.96 | 96.91 | 47.22 |
| Laplacian | 94.65 | 50.25 | 94.68 | 50.29 | 95.28 | 49.80 | 94.66 | 50.34 | 94.82 | 50.17 |
| Gaussian | 94.46 | 50.83 | 95.17 | 50.33 | 94.80 | 50.46 | 95.09 | 50.80 | 94.88 | 50.60 |
| Cosine (CoP) | 67.25 | 83.41 | 75.53 | 79.93 | 82.48 | 73.83 | 8.33 | 98.29 | 58.40 | 83.86 |
| Cosine-Polynomial | 54.10 | 84.48 | 75.97 | 75.04 | 82.82 | 69.01 | 59.15 | 83.27 | 68.01 | 77.95 |
| Cosine-Laplacian | 76.18 | 77.95 | 77.54 | 76.70 | 84.47 | 70.16 | 11.97 | 97.57 | 62.54 | 80.60 |
| Cosine-Gaussian (CoRP) | 50.07 | 89.32 | 62.56 | 83.74 | 72.76 | 78.91 | 9.02 | 98.14 | 48.60 | 87.53 |

- The cosine-polynomial kernel does not hold the $\ell_1$ nor $\ell_2$ distance-preserving property for $\phi_{\cos}(\boldsymbol{z})$-space, as the polynomial kernel is defined as:

$$k_{\text{poly}}(\boldsymbol{z}_1, \boldsymbol{z}_2) = (\boldsymbol{z}_1^\top \boldsymbol{z}_2 + c)^d. \tag{17}$$

  To obtain an explicit feature mapping for $k_{\text{poly}}$, we do not adopt the RFFs and take the Tensor Sketch approximation [56] instead for simplicity.

Table 6 illustrates the comparisons on the detection performance among multiple alternative kernels. Actually, both the cosine-Laplacian kernel and the cosine-polynomial kernel cannot bring detection performance gains on top of the cosine kernel (CoP), which indicates that the $\ell_1$-distance relationship characterized by the Laplacian kernel and the inner-product information characterized by the polynomial kernel in the $\phi_{\cos}(\boldsymbol{z})$-space are less effective in promoting the separability between InD and OoD features. Thus, the cosine-Gaussian kernel is used and recommended in the proposed KPCA method for OoD detection.

## C.3 Sensitivity analysis

A comprehensive sensitivity analysis is executed to show the effects of hyper-parameters in CoP and CoRP. A common hyper-parameter in CoP and CoRP is the number of columns $q$ of the dimensionality-reduction matrix $\boldsymbol{U}_q^\Phi$. Additional hyper-parameters of CoRP include the bandwidth $\gamma$ of the Gaussian kernel and the number of RFFs $M$. In the following, we discuss the influence of each hyper-parameter, and report experiment results of the detection performance by varying one hyper-parameter with the others fixed.

**Effect of $q$** $q$ indicates the number of preserved principal components and determines how much information captured by the subspace where the InD and OoD data is projected onto. $q$ is selected as the minimal number of principal components with the amount of information that exceeds the given explained variance ratio.

Figure 5 illustrates the detection performance of CoP and CoRP under varied explained variance ratios. On CIFAR10 and ImageNet-1K benchmarks, for CoP, a mild value of the explained variance ratio is suggested with around 90% for keeping the components. Regarding CoRP, a sufficiently large value of the explained variance ratio is no longer essential for CoRP on the ImageNet-1K benchmark, which might be due to that the 2 concatenated kernels make the useful information for distinguishing OoD samples more concentrated in less principal components.

**Effect of $\gamma$** The Gaussian kernel width $\gamma$ directly affects the mapped data distribution. For a large $\gamma$, $k_{\text{gau}}(\boldsymbol{z}_1, \boldsymbol{z}_2) = e^{-\gamma\|\boldsymbol{z}_1 - \boldsymbol{z}_2\|_2^2} \approx 0$ for $\boldsymbol{z}_1 \neq \boldsymbol{z}_2$, which indicates that the mapped features of $\boldsymbol{z}_1$ and $\boldsymbol{z}_2$ are (nearly) mutually-orthogonal. In this case, a PCA would become meaningless. For a small $\gamma$, the KPCA-based reconstruction errors will approach the standard PCA-based ones, shown by [57]. Figure 6 illustrates the detection FPR95 and AUROC values of CoRP *w.r.t* varied Gaussian kernel width $\gamma$ on CIFAR10 and ImageNet-1K benchmarks. Clearly, neither a too large nor a too small kernel width benefits the detection performance, and a mild value of $\gamma$ should be carefully tuned for different in-distribution data.

**Effect of $M$** The number of RFFs $M$ determines the approximation ability of RFFs towards the Gaussian kernel. As proved in [13], the larger the $M$, the better the RFFs approximate $k_{\text{gau}}$.

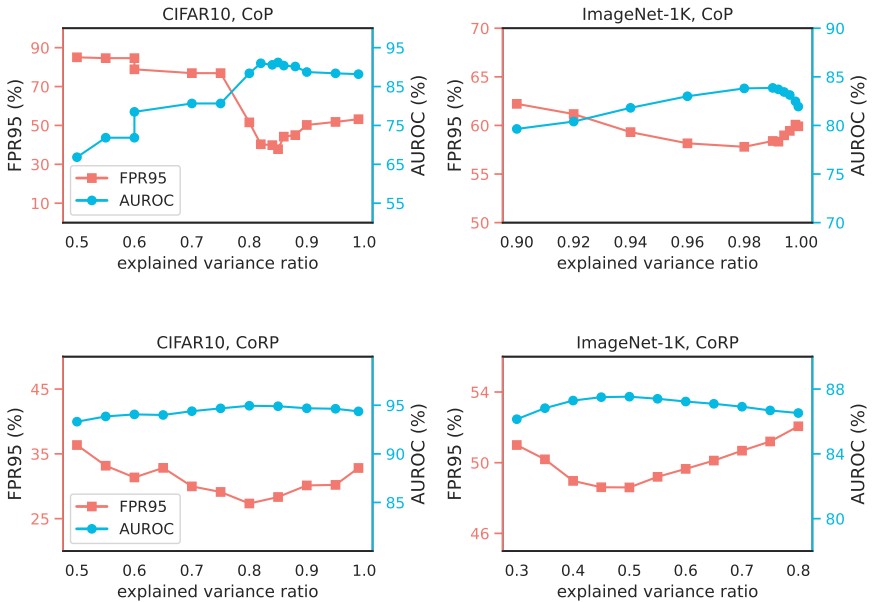

Figure 5: A sensitivity analysis on the explained variance ratio of CoP (top) and CoRP (bottom). The average FPR and AUROC values of OoD data sets in CIFAR10 and ImageNet-1K benchmarks are reported. The Gaussian kernel width $\gamma$ and the dimension $M$ of RFFs in CoRP are fixed.

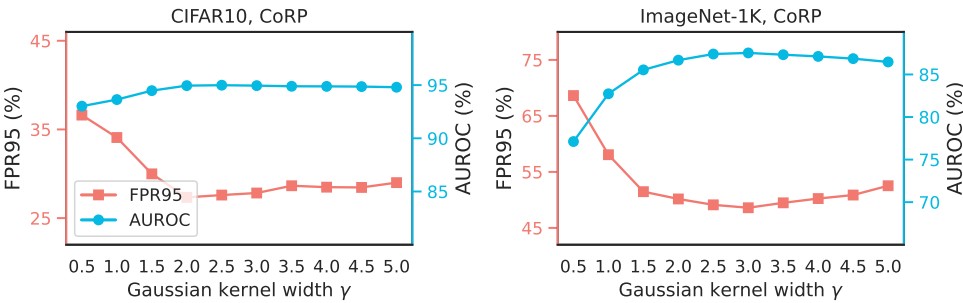

Figure 6: A sensitivity analysis on the Gaussian kernel width $\gamma$ of CoRP. The average detection FPR and AUROC values of OoD data sets in CIFAR10 and ImageNet-1K benchmarks are reported. The explained variance ratio and the dimension $M$ of RFFs are fixed.

Figure 7 indicates the detection FPR95 and AUROC values of CoRP *w.r.t.* multiple values of the RFFs dimension $M$ on CIFAR10 and ImageNet-1K benchmarks. As $M$ increases, the detection performance gets improved since the RFFs better converge to the Gaussian kernel. Considering the computation efficiency of eigendecomposition on the covariance matrix of $\mathbb{R}^{M \times M}$, in the comparison experiments, we adopt $M = 4m$ on CIFAR10 with $m = 512$ for ResNet18, and $M = 2m$ on ImageNet-1K with $m = 2048$ for ResNet50 and $m = 1280$ for MobileNet.

# D    Supplementary theoretical results

The proof of Proposition 1 is presented.

*Proof.* Recall $\boldsymbol{z} \in \mathbb{R}^m$ and suppose $\Phi : \mathbb{R}^m \to \mathbb{R}^M$ and $\boldsymbol{U}^\Phi \in \mathbb{R}^{M \times M}$ is the eigenvector matrix of the covariance matrix of the training data with $\boldsymbol{U}^\Phi = \left[ \boldsymbol{U}_q^\Phi, \boldsymbol{U}_p^\Phi \right]$ and $q + p = M$. For the reconstruction error $e^\Phi(\hat{\boldsymbol{z}})$ of a new test sample $\hat{\boldsymbol{z}} \in \mathbb{R}^m$ in the mapped $\Phi(\boldsymbol{z})$-space, we have:

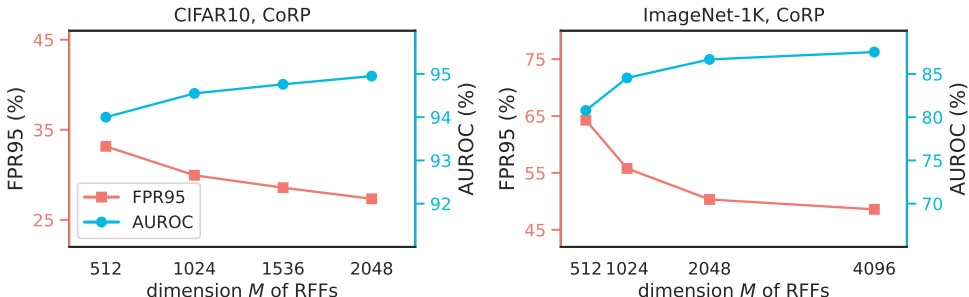

Figure 7: A sensitivity analysis on the dimension $M$ of RFFs of CoRP. The average detection FPR and AUROC values of OoD data sets in CIFAR10 and ImageNet-1K benchmarks are reported. The explained variance ratio and the Gaussian kernel width $\gamma$ are fixed.

$$
\begin{aligned}
\left(e^{\Phi}(\hat{\boldsymbol{z}})\right)^2 &= \left\| (\Phi(\hat{\boldsymbol{z}}) - \boldsymbol{\mu}^{\Phi}) - \boldsymbol{U}_q^{\Phi} \boldsymbol{U}_q^{\Phi\top} (\Phi(\hat{\boldsymbol{z}}) - \boldsymbol{\mu}^{\Phi}) \right\|_2^2 \\
&= \left( (\Phi(\hat{\boldsymbol{z}}) - \boldsymbol{\mu}^{\Phi}) - \boldsymbol{U}_q^{\Phi} \boldsymbol{U}_q^{\Phi\top} (\Phi(\hat{\boldsymbol{z}}) - \boldsymbol{\mu}^{\Phi}) \right)^{\top} \left( (\Phi(\hat{\boldsymbol{z}}) - \boldsymbol{\mu}^{\Phi}) - \boldsymbol{U}_q^{\Phi} \boldsymbol{U}_q^{\Phi\top} (\Phi(\hat{\boldsymbol{z}}) - \boldsymbol{\mu}^{\Phi}) \right) \\
&= (\Phi(\hat{\boldsymbol{z}}) - \boldsymbol{\mu}^{\Phi})^{\top} (\Phi(\hat{\boldsymbol{z}}) - \boldsymbol{\mu}^{\Phi}) - (\Phi(\hat{\boldsymbol{z}}) - \boldsymbol{\mu}^{\Phi})^{\top} \boldsymbol{U}_q^{\Phi} \boldsymbol{U}_q^{\Phi\top} (\Phi(\hat{\boldsymbol{z}}) - \boldsymbol{\mu}^{\Phi}) \\
&= (\Phi(\hat{\boldsymbol{z}}) - \boldsymbol{\mu}^{\Phi})^{\top} \left( \boldsymbol{I} - \boldsymbol{U}_q^{\Phi} \boldsymbol{U}_q^{\Phi\top} \right) (\Phi(\hat{\boldsymbol{z}}) - \boldsymbol{\mu}^{\Phi}) \\
&= (\Phi(\hat{\boldsymbol{z}}) - \boldsymbol{\mu}^{\Phi})^{\top} \boldsymbol{U}_p^{\Phi} \boldsymbol{U}_p^{\Phi\top} (\Phi(\hat{\boldsymbol{z}}) - \boldsymbol{\mu}^{\Phi}) \\
&= \left\| \boldsymbol{U}_p^{\Phi\top} (\Phi(\hat{\boldsymbol{z}}) - \boldsymbol{\mu}^{\Phi}) \right\|_2^2 .
\end{aligned}
\tag{18}
$$

Obviously $e^{\Phi}(\hat{\boldsymbol{z}}) = \left\| \boldsymbol{U}_p^{\Phi\top} (\Phi(\hat{\boldsymbol{z}}) - \boldsymbol{\mu}^{\Phi}) \right\|_2$ and the proof is finished. $\qquad\square$

The key in the proof of Proposition 1 is $\boldsymbol{U}_q^{\Phi} \boldsymbol{U}_q^{\Phi\top} + \boldsymbol{U}_p^{\Phi} \boldsymbol{U}_p^{\Phi\top} = \boldsymbol{I}$. Since $\boldsymbol{U}^{\Phi}$ is the eigenvector matrix of the covariance matrix, thereby $\boldsymbol{U}^{\Phi}$ is a unitary matrix and satisfies $\boldsymbol{U}^{\Phi} \boldsymbol{U}^{\Phi\top} = \boldsymbol{U}^{\Phi\top} \boldsymbol{U}^{\Phi} = \boldsymbol{I}$, which leads to $\boldsymbol{U}_q^{\Phi} \boldsymbol{U}_q^{\Phi\top} + \boldsymbol{U}_p^{\Phi} \boldsymbol{U}_p^{\Phi\top} = \boldsymbol{I}$.

