# OpenReview forum: "Kernel PCA for Out-of-Distribution Detection"
_NeurIPS.cc/2024/Conference — NeurIPS 2024 poster_

### Official Review · Reviewer_6eWv · 2024-07-08

**Soundness:** 3
**Presentation:** 3
**Contribution:** 3
**Rating:** 6
**Confidence:** 3

**Summary:**

The paper describes a method for out-of-distribution (OOD) detection. Kernel PCA is applied to the penultimate features of a neural network. Two kernels are chosen motivated by prior work on nearest-neighbour-based detection. Namely the cosine kernel and a cosine-Gaussian kernel. In experiments, both methods outperform baseline methods on image-based OOD detection.

**Strengths:**

- The paper is well-written and easy to follow. The work is put into context, clearly motivated and the methods are explained clearly.
- The method itself is fairly simple and works well which is highly appreciated.
- Extensive experiments and comparisons with prior work are conducted to showcase the method's performance.

**Weaknesses:**

- in section 2 it is not entirely clear whether the RFF approximation is used for the the cosine-Gaussian kernel alone or also for the cosine kernel. Also please, elaborate on the details of how the kernel is approximated.

- in relation to the last point, in the experiments, it is important to highlight the dimensionality of the penultimate layer features $c$ and the number of random Fourier features that you select. It is known that for medium to high-dimensional problems, the approximation with RFF requires a very high number of features.

- line 198: the computational complexity of the RFF is not discussed in this paragraph.

- section 2.2 is less related work and more background. Thus, sections 2.2 and 3.1 could be better summarized in a "background" section.

- It is not entirely clear how Proposition 2 is necessary to implement CoP and CoRP. Please clarify this in the manuscript.

- at the start of the paper it is unclear why you two kernels (lines 65, 68, ...). Only from the experiments and algorithm 1, it becomes clear that these are two different choices/methods.

**Questions:**

- it is unclear how the proposed methods achieve O(1) time and memory complexity with RFF. Please elaborate on this.
- line 133: how is $q$ chosen?
- in Figure 2, please clarify what the kernel function in this experiment exactly is.

Smaller things:
- line 282: here the standard solution from [1] should be mentioned
- line 192: wrap up
- line 259: "under the same framework" unclear phrasing
- line 299: "we adopt" remove "should"
- for automatic learning of features from data, a suggestion is to consider Recursive Feature Machines [2].

References:
[1] G. H. Bakır, J. Weston, and B. Schölkopf, “Learning to find pre-images,” in Proc. Adv. Neural Inf. Process. Syst., 2004, pp. 449–456.

[2] A. Radhakrishnan et al., Mechanism for feature learning in neural networks and backpropagation-free machine learning models. Science 383, 1461-1467 (2024).

**Limitations:**

For reproducibility the main limitation lies in the missing information about the RFF implementation as no information is provided about the number of random Fourier features. Furthermore, it is not clear what other kernel hyperparameters are to be tuned and how they are selected.

---

> ### Author Rebuttal · Authors · 2024-08-06
>
> Thanks for the detailed comments, each of which will be answered pointwisely below.
> Due to length limit, we
> - use "W", "Q" for Weakness, Question,
> - put the references in global response,
> - put the mentioned figures in the one-page rebuttal PDF file.
>
> ---
> ### **(W1) The usage of RFF approximation**
> - For the *Cosine* kernel, the RFF approximation is *not* needed, since the Cosine kernel $k_{\rm cos}(z_1,z_2)=\frac{z_1^\top z_2}{||z_1||\_2||z_2||\_2}$ has its exact feature mapping $\phi_{\rm cos}(z)=\frac{z}{||z||\_2}$, such that $k_{\rm cos}(z_1,z_2)=\phi_{\rm cos}(z_1)^\top\phi_{\rm cos}(z_2)$.
> - For the *Cosine-Gaussian* kernel, _the RFF is adopted to only approximate the Gaussian kernel part_, i.e.,  $k_{\rm gau}(z_1,z_2)\approx\phi_{\rm RFF}(z_1)^\top\phi_{\rm RFF}(z_2)$ in Eqn.(2) in the paper. Specifically, the adopted feature mapping $\phi_{\rm RFF}(\phi_{\rm cos}(z))$ for the entire *Cosine-Gaussian* kernel is a composite of the exact feature mapping $\phi_{\rm cos}$ for the Cosine kernel part and the approximate RFF mapping $\phi_{\rm RFF}$ for the Gaussian kernel $k_{\rm gau}$ part.
>
> ---
> ### **(W2) Dimensionality of penultimate layer features $m$ and the number of RFFs $M$**
> The settings of  $m$ and $M$  are provided in **Table 6eWv-1**, where $M$ is  set as $4m$ for the CIFAR10 InD and $2m$ for the ImageNet-1K InD for all experiments in the paper.
>
> **Table 6eWv-1** *The values of $m$ and $M$*
> dataset|network|feat. dim. $m$|num. of RFFs $M$
> :-:|:-:|:-:|:-:|
> CIFAR10|ResNet18|512|2048
> ImageNet-1K|ResNet50|2048|4096
> ImageNet-1K|MobileNet|1280|2560
>
> Indeed, as $M$ increases, RFF can better approximate the Gaussian kernel. We have conducted sensitivity analysis on $M$ in **line 605 and Fig.9 of Appendix D.3**, showing that increasing $M$ boosts and then maintains rather steady OoD detection performances. Hence, we note that _a substantially high number of RFF is not of significant necessity in the addressed task_. To balance efficiency and effectiveness, we adopt $M$ as in **Table 6eWv-1** with SOTA results under varied setups.
>
> ---
> ### **(W3) Computation complexity of RFF**
> With Eqn.(2), for the RFF mapping $\phi_{\rm RFF}$, $2M$ samplings for $\omega_i$ and $u_i$ respectively are required, and $\phi_{\rm RFF}$ includes $M$ dot products and $M$ additions. Accordingly, the memory and time complexity of RFF is ${\cal O}(M)$.
>
> ---
> ### **(W4) Organization of Sec.2.2 and Sec.3.1**
> Thanks for the nice suggestion. We will merge Sec.2.2 and Sec.3.1 as a 'background' section in the revised version.
>
> ---
> ### **(W5) Proposition 2 and the implementation of CoP and CoRP**
> Proposition 2 supplements the analysis for kernel implementation of CoP and CoRP.
>
> CoP and CoRP leverage the covariance-based KPCA and are implemented via explicit feature mappings induced from the kernel, rather than via the more expensive kernel implementation. Proposition 2 provides the reconstruction errors when implemented with kernels, and serves as a supplementary analysis on our more efficient covariance-based CoP and CoRP.
>
> ---
> ### **(W6) Elaboration on the two kernels**
> The proposed OoD detector is a KPCA-based method with **two alternative choices of kernels** explained in the main context, i.e., Cosine kernel and Cosine-Gaussian kernel.
>
> ---
> ### **(Q1) How O(1) complexity is achieved with RFF**
> Thanks for the valuable comment. To align with the complexity analysis of KNN detector [7], the O(1) time and memory complexity in our work also refers to the inference procedure of calculating the OoD detection score, i.e., the reconstruction error, when given with features, so it indeed does not include the computation of the mapping.
> Regarding the raised comment in **W3**, we specify the complete computation complexity of CoP and CoRP:
> - For CoP without RFF involved, the time and memory complexity in inference is  O(1), since the feature mapping $\phi_{\rm cos}$ is obtained right away as the normalized $z$ without additional cost.
> - For CoRP with RFF involved, when considering the computation of the feature mapping  $\phi_{\rm RFF}(\phi_{\rm cos}(\cdot))$, we have the complete time and memory complexity in inference as ${\cal O}(M)$, since $\phi_{\rm RFF}$ requires an ${\cal O}(M)$ complexity.
>
> ---
> ### **(Q2) How to choose $q$**
> In CoP and CoRP, $q$ is selected as the number of principal components sufficing a certain explained variance ratio (e.g., 99%), which is also a common technique in (K)PCA-based methods. For example, given the ResNet50 on ImageNet-1K with a penultimate feature dimension $m=2048$, to preserve an explained variance ratio of 99%,  $q=1346$ principal components are required in CoP.
>
> We have conducted sensitivity analysis on $q$ in CoP and CoRP to comprehensively evaluate its effect. More details can refer to **line 587 and Fig.7 in Appendix D.3**.
>
> ---
> ### **(Q3) Kernel function in Fig.2**
> In the left panel of Fig.2, the adopted kernel function is the Cosine kernel function $k_{\rm cos}$. In the right panel of Fig.2, the adopted kernel function is the Gaussian function $k_{\rm gau}$ on $\ell_2$-normalized features, i.e., Cosine-Gaussian kernel.
>
> ---
> ### **(Q4) Smaller things**
> - **Typos and literatures.** We will revise all the typos and add the recommended literature [9] in the updated version. Also, we will consider the suggested Recursive Feature Machines [10] in learning algorithms for OoD detection in future works.
> - **Unclear phrasing.** Regarding the 'under the same framework' in line 259, it indicates that our KPCA reconstruction error and the regularized reconstruction error in [6] are evaluated under the same fusion trick as [6] for a fair comparison.
>
> ---
> ### **Limitation on reproducibility**
> The source code for reproduction has been provided in the supplementary material, and will also be released publicly in github upon acceptance. In **Appendix D.3**, all  hyper-parameters of CoP and CoRP are discussed in detail and evaluated with ablation studies.

---

> > ### Comment · Reviewer_6eWv · 2024-08-09
> > **Official comment by reviewer 6eWv**
> >
> > I would like to thank the authors for their responses. My questions have been addressed.
> >
> > The last remaining point is to ask the authors, to phrase their manuscript such that a clear distinction between CoP and CoRP is made from the beginning. This regards the use of RFF, the complexity and that these are two distinct contributions. Thus, claims about the complexity, such as in the abstract, need to be clarified (and rectified) with the $\mathcal{O}(M)$ complexity.

---

> > > ### Author Response · Authors · 2024-08-10
> > >
> > > Dear Reviewer,
> > >
> > > We agree with your suggestions. In the final version, we will polish up our manuscript with elaboration and clarification on
> > > - the distinction between CoP and CoRP in the beginning of introducing our method and also in abstract,
> > > - the use of RFF in Gaussian kernel approximation in CoRP and also the corresponding two distinctive contributions of our kernel designs specified for OoD detection,
> > > - the clarified computation complexity of CoRP as ${\mathcal O}(M)$ with RFF as explained in our above rebuttal,
> > >
> > > which are viable to be included into the final version.
> > >
> > > We thank the Reviewer for recognizing our rebuttal responses and the helpful advice. We would appreciate if our responses are helpful for the final evaluations on our work.
> > >
> > > Best regards,
> > > Authors

---

> > > > ### Comment · Reviewer_6eWv · 2024-08-12
> > > > **Official comment by reviewer 6eWv**
> > > >
> > > > Thanks, this answers all my concerns and I have updated my score accordingly.

---

### Official Review · Reviewer_w1ZW · 2024-07-09

**Soundness:** 3
**Presentation:** 3
**Contribution:** 3
**Rating:** 5
**Confidence:** 4

**Summary:**

The paper everage the framework of Kernel PCA (KPCA) for OoD detection, and seek suitable non-linear kernels that advocate the separability between InD and OoD data in the subspace spanned by the principal components. Besides, explicit feature mappings induced from the devoted task specific kernels are adopted so that the KPCA reconstruction error for new test samples can be efficiently obtained within an O(1) time and memory complexity in inference.

**Strengths:**

1. The paper is written well and is easy to understand.

2. The studied problem is very important.

3. The results seem to outperform state-of-the-art.

**Weaknesses:**

1. It might be more useful to include more effective post-hoc OOD detection methods, such as ReAct, ASH for comparison. Also, the training-based approach might be useful to be included.

2. It might be more useful to include the ablation results on more kernel designs.

3. I am wondering why linear separable space between ID and OOD matters, if we have a non-linear classifier, it can still perform accurate OOD detection.

4. What are some important assumptions here for the theory? Might be useful to provide some justifications if there are any.

**Questions:**

see above

---

> ### Author Rebuttal · Authors · 2024-08-06
>
> Thanks for the detailed comments, each of which will be answered pointwisely below.
> Due to length limit, we
> - use "W", "Q", "L" for Weakness, Question, and Limitation,
> - put the references in global response,
> - put the mentioned figures in the one-page rebuttal PDF file.
>
> ---
> ### **(W1) More comparisons**
> As suggested, we add the post-hoc method ASH [3] and a training-based method DML [4] for more comprehensive comparisons, shown in **Table w1ZW-1**. Another suggested method ReAct [5] has been considered into comparisons in **Table 3** and **Table 5**. Besides, we also supplement two latest post-hoc methods SCALE [1] and DDCS [2], as suggested by *Reviewer sSKR*.
>
> **Table w1ZW-1** *Additional comparisons with more latest methods with ResNet50 on ImageNet-1K as InD data. Our CoRP detector is set upon ReAct [5,6] similar to Table 3 in the paper.*
> method|iNaturalist|SUN|Places|Textures|AVERAGE-FPR ($\downarrow$)
> :-:|:-:|:-:|:-:|:-:|:-:|
> ASH-P|44.57|52.88|61.79|42.06|50.32
> ASH-B|14.21|22.08|33.45|21.17|22.73
> ASH-S|11.49|27.98|39.78|11.93|22.80
> DML|47.32|57.40|61.43|52.80|54.74
> DML+|13.57|30.21|39.06|36.31|29.79
> SCALE|9.50|23.27|34.51|12.93|20.05
> DDCS|11.63|18.63|28.78|18.40|19.36
> CoRP+ (ours)|10.77|18.70|28.69|12.57|**17.68**
>
> ---
> ### **(W2) Ablation on more kernel designs**
> In our work, we have demonstrated the indispensable importance of the $\ell_2$-normalization feature mapping from Cosine kernel. On such basis, upon the Cosine kernel, we further deploy a Gaussian kernel to pursue greater separability between InD and OoD, where the $\ell_2$-distance along samples are preserved: $k_{\rm gau}(z_1,z_2)=e^{-\gamma||z_1-z_2||_2}$. Further investigations on more kernel designs are shown as follows. More details can also be found in **Appendix D.1 and D.2**.
> - **Cosine-Laplacian kernel.** Aside from the Gaussian kernel preserving the $\ell_2$-distance on top on the indispensable Cosine kernel, we have considered another *Cosine-Laplacian* kernel, where the Laplacian kernel could keep the $\ell_1$-distance: $k_{\rm lap}(z_1,z_2)=e^{-\gamma||z_1-z_2||_1}$. Comparisons in **Table w1ZW-2** demonstrate the superiority of Gaussian kernel, and refer to **Fig.6** of Appendix D.2 for thorough results.
> - **Cosine-polynomial kernel.** The polynomial kernel $k_{\rm poly}(z_1,z_2)=(z_1^\top z_2+c)^d$ is also investigated via the Tensor Sketch approximation [8] in **Table w1ZW-2**, and is shown to be unable to bring superior detection performances.
> - **Effects of individual kernels.** We have also investigated the individual effect of each kernel in **Table w1ZW-2**, where the $\ell_2$-normalization mapping from Cosine kernel shows indispensable significance for OoD detection and the preserved $\ell_2$-distance from Cosine-Gaussian kernel brings the best results. More thorough results can refer to **Fig.5** of Appendix D.1.
>
> **Table w1ZW-2** *Comparisons among different kernels with ResNet18 on CIFAR10 as InD data.*
> kernel|SVHN|LSUN|iSUN|Textures|Places365|AVERAGE-FPR ($\downarrow$)
> :-:|:-:|:-:|:-:|:-:|:-:|:-:|
> Cosine (CoP)|11.56|23.24|53.71|26.28|74.11|37.78
> polynomial|46.80|97.43|95.25|61.97|95.91|79.47
> Laplacian|46.63|72.65|73.00|49.15|70.12|62.31
> Gaussian|94.93|95.09|94.53|94.49|94.78|94.76
> Cosine-polynomial|26.89|37.01|50.32|33.16|76.95|44.86
> Cosine-Laplacian|37.80|25.58|42.20|31.05|57.28|38.78
> Cosine-Gaussian (CoRP)|20.68|19.19|21.49|21.61|53.73|**27.34**
>
> ---
> ### **(W3) Why not non-linear classifier & why linear separability matters**
> We would like to address this point from the following two aspects:
> - **Non-linear classifier is impractical for general OoD detection tasks.** OoD detection is an *unsupervised* task, as the DNN is trained on InD data without knowledge about OoD data. In OoD detection, OoD data can be any data of any size drawn from different distributions w.r.t InD. Thus, it is impractical to particularly train a non-linear binary classifier with sufficient OoD data being another class. Then, it is justified to consider *unsupervised* models that brings separability between InD and OoD, such as the KNN-based detector in [7], and our KPCA-based detector.
> - **Linearity separability matters.** In our work, we find that the linear principal components by simply applying PCA  are insufficient to depict the informative patterns of InD features,  as InD features are not even located compactly nor linearly inseparable with OoD features, as shown in **Fig.1(a)**. Thus, PCA cannot achieve good reconstruction performances for all InD features, so that the reconstruction of InD and OoD features along such extracted principal components from linear PCA can both be bad, failing to well differentiate OoD from InD.
>
>     Therefore, we introduce non-linear kernels with KPCA which map InD and OoD features into a new space, where the corresponding InD mappings are located more compactly and are almost linearly separable from OoD mappings. Thanks to such separability between InD and OoD, informative principal components are learned for InD and only works for InD with good reconstruction, as the separable OoD fails to be well characterized along such principal components, as shown in **Fig.1(b)**.   In this sense, good OoD detection performances are attained through the reconstructions along new axes by the extracted components in the mapped feature space with KPCA.
>
> ---
> ### **(W4) Important assumptions for the theory**
> Important assumptions for the theory in the paper are listed.
> - For the RFF approximation outlined in Sec.2.2, the theory basis of RFF methods is **the Bochner's theorem**, which assumes *a continuous and shift-invariant kernel* (**line 118**). In our work, we apply RFF to approximate the Gaussian and Laplacian kernels, both of which satisfy the assumption in the Bochner's theorem.
> - For the analytical analysis in Sec.5, the two propositions have no particular assumptions, and can be applied to any standard KPCA methods under both covariance and kernel setups.

---

> ### Author Response · Authors · 2024-08-12
>
> Dear Reviewer,
>
> We hope that our responses above could address your concerns and be helpful for final evaluations on our work.
>
> As the author-reviewer discussion is approaching the end, we would like to inquiry  if there is any question, and we would be willing to discuss.
>
> Sincerely, Authors.

---

### Official Review · Reviewer_LuQe · 2024-07-12

**Soundness:** 3
**Presentation:** 3
**Contribution:** 2
**Rating:** 5
**Confidence:** 4

**Summary:**

The paper introduces a method that applies Kernel Principal Component Analysis (KPCA) to the output of the penultimate layer of a Deep Neural Network (DNN) for out-of-distribution (OOD) detection. Inspired by previous approaches in OOD detection that utilize K-nearest neighbors (KNN) on $\ell_{2}$ normalized features, two kernels are designed to incorporate these features. The study explores the effectiveness of these kernels and discusses experimental results.

**Strengths:**

1. The paper is well-written and easy to follow.

2. The problem of OOD detection is important in ML and relevant to the Neurips community.

3. Many experiments are conducted and some are promising. For instance, the proposed approach using the Cosine kernel and Gaussian-Cosine kernel performs better than KNN work [1] which the method is inspired.

**Weaknesses:**

1. It is not clear how the proposed method can adapt to a wider variety of feature maps and effectively leverage corresponding kernels. The current feature map used in this study draws inspiration from [1] and closely resembles it and the main observation of the paper is to consider to corresponding kernel to it. As a result, the effectiveness and novelty of the kernel method in this context are not entirely clear and need to be clarified.

2. The authors mention ‘kernel learning’ as a potential area for future exploration under the study's limitations. However, pursuing kernel learning in this context may appear redundant given the extensive training already performed on the DNN from scratch for the OOD detection task.

[1]Yiyou Sun, Yifei Ming, Xiaojin Zhu, and Yixuan Li. Out-of-distribution detection with deep 358 nearest neighbors. In International Conference on Machine Learning, pages 20827–20840. 359 PMLR, 2022.

**Questions:**

1. How can one select a kernel for OOD detection beyond those discussed in this work? Please refer to my previous comment as well.

2. Could you provide a mathematical justification for why KPCA with a Gaussian cosine kernel or a cosine kernel itself performs better than the KNN method with $\ell_{2}$ normalization, as discussed in [1]?

[1]Yiyou Sun, Yifei Ming, Xiaojin Zhu, and Yixuan Li. Out-of-distribution detection with deep 358 nearest neighbors. In International Conference on Machine Learning, pages 20827–20840. 359 PMLR, 2022.

**Limitations:**

The authors discussed the limitations of their work.

---

> ### Author Rebuttal · Authors · 2024-08-06
>
> Thanks for the detailed comments, each of which will be answered pointwisely below.
>
> ---
> ### **(Weakness 1 & Question 1) Selection of kernels/feature mappings**
> We would like to firstly elaborate the effectiveness and novelty of our KPCA method (*Weakness 1*), and then discuss more kernel selections (*Question 1*).
>
> **(A1) Novelty of KPCA with the designed two effective kernels for OoD detection**
> - The key of our KPCA detector is to find qualified kernels (or their explicit mappings) that can capture informative patterns of InD and meanwhile well distinguish InD and OoD in the mapped space.
> From our extensive evaluations, we have highlighted that: *the generic KPCA method can adapt to a wide variety of kernels, but only specific designs of kernels are competent for OoD detection*, which is justified in detail in **A2** below. Thus, although the KPCA detector is not entirely new in the sense of kernel methods, *we have brought new insights into OoD detection from a kernel perspective, which is novel to the research community and the first trial on practicable and efficacious kernel methods for OoD detection*.
> - The efficiency of our KPCA detector is underlined for its cheap O(1) complexity from the reconstruction-error score in OoD inference over the O(N) of KNN. Besides, explicit feature mappings instead of expensive kernel matrix computation are deployed in our detector, resolving the inefficiency of KPCA on large-scale data.
> - Despite the used $\ell_2$-normalization technique, our method is significantly different from KNN, which will be discussed in the response to **Question 2**.
>
> **(A2) Investigations on more kernel choices**
> The devoted two kernels provide the following in-depth insights about InD and OoD that should be considered in designing more viable kernels:
> - The $\ell_2$-normalization from Cosine kernel plays a pivotal role in distinguishing OoD from InD by balancing the norm of InD and OoD features.
> - The $\ell_2$-distance preserving property of the shift invariant Gaussian kernel can further promote the separability that benefits OoD detection performances.
>
> Therefore, when devising or learning new kernels, we should consider those kernels that can normalize feature norms or preserve the distance relationships. For example, the ablation studies in **Fig.6** of **Appendix D.2** discuss an alternative **cosine-Laplacian kernel**, where the $\ell_1$-distance gets preserved via the Laplacian kernel. Another common **polynomial** kernel is also investigated in **Table LuQe-1**.
>
> **Table LuQe-1** *More kernel choices with ResNet18 on CIFAR10 as InD.*
> kernel|AVERAGE-FPR ($\downarrow$)
> :-:|:-:|
> Cosine (CoP)|37.78
> polynomial|79.47
> Laplacian|62.31
> Gaussian|94.76
> Cosine-polynomial|44.86
> Cosine-Laplacian|38.78
> Cosine-Gaussian (CoRP)|**27.34**
>
>
> ---
> ### **(Weakness 2) Limitation of learning kernels as future works**
> Learning kernels as a potential area may not be redundant for OoD detection.
> - Despite the training on DNNs from scratch, it is also a widely-accepted research paradigm to further perform learning with features from a pre-trained DNN. A typical example is the celebrated Deep Kernel Learning (DKL, [1]), where a kernel learning procedure is imposed upon the features learned from a pre-trained DNN.
> With particular regards to OoD detection, an additional learning step can be taken for the specified goal. Here, DKL could be considered as an alternative choice so as to pursue stronger kernels that can better characterize InD and OoD with enhanced detection performance. In contrast, we currently leverage the kernel method under typical non-parametric setups, rather than learning-based ones.
> - Although we leverage the non-parametric kernel method, it still requires a few hyper-parameters. In this work, we tune the parameters $q$, $M$ and $\gamma$ of our proposed CoP and CoRP and use the suggested default settings according to our extensive evaluations. In future work, it would be of greater convenience and potentials, if such kernels can be learned with data available at hands.
>
> [1] Wilson, et al. "Deep kernel learning." AISTATS'2016.
>
> ---
> ###  **(Question 2) Justification between KNN and KPCA**
> KNN relies on the discrete distance, while the analysis on KPCA comes from the feature space w.r.t Mercer kernel operators. Thus, it is difficult to align theoretical justifications between such two methods, nor to make comparisons straight away. Despite the $\ell_2$-normalization technique, our KPCA detector has significant differences with KNN, regarding the designed kernel tricks and O(1)-complexity detection score. With particular interest to OoD detection, the following two aspects are elaborated to discuss the superiority of our KPCA detector over the KNN one.
>
> **OoD detection score design.** KNN sets the Euclidean distance between samples as the detection score, while KPCA adopts the reconstruction error along the extracted principal components. For the former, the k-th smallest Euclidean distance to InD data  determines whether a sample is OoD or not; thus, only the nearest affinity neighborhood in InD data is used, where  the full information of InD is missing in calculating the detection score. In contrast, in the latter, all InD data contributes to learning the principal components for the reconstruction; accordingly, our KPCA-based method goes beyond the affinity neighborhood in KNN, but makes full use of InD data in designing the detection score, facilitating stronger OoD detection.
>
> **Low-dimensional subspace modeling.** Another key advantage of KPCA comes from the low-dimensional subspace learned from InD data, where the most informative patterns of InD data (e.g., 99% explained variance) is kept and redundant information in InD data is removed. Such a subspace is learned with specification to InD data and is thereby less sensitive in revealing the pattern of OoD data than using the original all dimensions of KNN. Hence, OoD data can get more easily differentiated.

---

> > ### Comment · Reviewer_LuQe · 2024-08-09
> >
> > Dear Authors,
> >
> > Thank you for your response. I have a follow-up question regarding your answer about extending to a more general class of kernels. Specifically, regarding computational complexity: as I understand it, efficient inference time is achieved only for the two specific kernels considered in your work (e.g., Cosine and Cosine-Gaussian kernels). For the Cosine kernel, the feature map has the same dimension as the feature space, while for the Cosine-Gaussian kernel, the feature map is approximated using Random Fourier Features (RFF).
> >
> > Could you comment on whether this efficient inference time can be achieved for other classes of kernels, such as those with potentially infinite-dimensional or high-dimensional feature spaces, beyond the two specific cases considered in your work?
> >
> > Sincerely,
> > Reviewer

---

> > > ### Author Response · Authors · 2024-08-10
> > >
> > > Dear Reviewer,
> > >
> > > Thanks for your reply. We agree with your understandings on the efficient inference time with the exact feature mapping and the approximated RFF mapping in our deployed Cosine and Cosine-Gaussian kernels, respectively.
> > >
> > > Regarding the extension to more general classes of kernels, our method can still achieve such efficient OoD inference, *as long as (the approximate) feature mappings w.r.t the kernel can  be obtained*, since the reconstruction error can be calculated straightly with an efficient ${\cal O}(1)$ complexity on mapped features. Several examples on explicit feature mappings of different types of kernels are discussed below.
> > > - The adopted *Gaussian* and *Laplacian* kernels are actually in correspondence to infinite-dimensional feature spaces and can be approximated via RFF, as in our work.
> > > - Another alternative is the *polynomial* kernel $k_{\rm poly}(z_1,z_2)=(z_1^\top z_2+c)^d$ w.r.t a high-dimensional space, which however does not satisfy the conditions of RFF. In this case, we can adopt relevant approximation methods for particular kernels, e.g., the Tensor Sketching in [1] for polynomial kernels as in **Table LuQe-1** in our rebuttal above.
> > > - A more general solution for generic kernels is provided: One can firstly use a sum of Gaussian kernels to approximate any kernel, and then adopt RFF to approximate the approximated Gaussian kernels, see an example in [2].
> > >
> > > Given these approximation methods, we would like to further highlight that the required dimension of the mapped features is not very high to ensure good OoD detection performances. Take the Gaussian kernel with infinite-dimensional spaces as an example, the dimension of approximated RFFs $M$ are set as $M=4m$ for the CIFAR10 InD and $M=2m$ for the ImageNet-1K InD throughout the paper, where $m$ is the dimension of DNN backbone features. Please kindly refer to **Fig.9** in our attached PDF in rebuttal and **Appendix D.3**, and our response **(W2)** and **Table 6eWv-1** to *Reviewer 6eWv*.
> > >
> > > We hope the responses above could well resolve your concerns and help the final evaluations on our work. If there is any further comment, we would like to discuss.
> > >
> > > Sincerely, Authors
> > >
> > > [1] Pham, et al. Fast and scalable polynomial kernels via explicit feature maps. KDD'2013.
> > > [2] Pennington et al. Spherical random features for polynomial kernels. NeurIPS'2015.

---

> > > > ### Comment · Reviewer_LuQe · 2024-08-11
> > > >
> > > > Dear Authors,
> > > >
> > > > Thank you for the rebuttal and for addressing my questions.
> > > >
> > > > The interesting aspect of the paper is that the KPCA perspective can reduce the computational cost of [1] while achieving comparable performance. I think this would be of interest to the OOD detection community. Additionally, the extensive experiments and evaluations provided are helpful.
> > > >
> > > > On the downside, I am still not convinced how your approach can be extended to a broader class of kernels while maintaining the computational advantage and/or providing new scores for OOD detection beyond the two kernels discussed. For example, if one uses the associated Gram matrix (which is common in KPCA), the computational complexity advantage of your method may be lost. Furthermore, it is not clear a priori how to select an appropriate kernel for the OOD detection task. However, this aspect could be explored more thoroughly in future work.
> > > >
> > > > Overall, I would like to maintain my score and vote to accept the paper. I would suggest to please discuss the limitations of your approach and the scope of your contributions in the abstract and introduction as mentioned in the above and in other reviews.
> > > >
> > > > Thank you,
> > > > Reviewer
> > > >
> > > > [1] Yiyou Sun, Yifei Ming, Xiaojin Zhu, and Yixuan Li. Out-of-distribution detection with deep nearest neighbors. In International Conference on Machine Learning, pages 20827–20840. PMLR, 2022.

---

> > > > > ### Author Response · Authors · 2024-08-11
> > > > >
> > > > > Dear Reviewer,
> > > > >
> > > > > We would like to thank you for recognizing our method being interesting to the OoD detection community, and your support on the acceptance of our work.
> > > > >
> > > > > Your suggestions on future works are inspirational to further explorations with kernel methods for OoD detection:
> > > > > - **Computation advantage on a broader class of kernels.** Our detection method is intentionally devised as a KPCA via *finite-dimensional (approximate) explicit feature mappings*,  which *avoids the exact computation on the Gram (kernel) matrix*, guaranteeing the efficient inference time. If only an arbitrary Gram/kernel matrix is available, it remains an interesting topic to be explored for efficient OoD detection with our KPCA-based method, as an (approximate) explicit feature mapping w.r.t the Gram (kernel) matrix is required with acceptable dimensions.
> > > > > - **Priori on more viable kernels.** In our work, we are grounded on the $\ell_2$-normalization property from the Cosine kernel for InD features and the $\ell_2$ distance-preserving property from the Gaussian kernel, jointly contributing to effective OoD detector with KPCA. Aside of the kernels discussed in rebuttal  **(A2)** above, we agree that a even broader class of kernels and more specific priori would be of interest  in future work for further understandings on kernel methods for OoD detection tasks.
> > > > >
> > > > > In the final version, as suggested, we will incorporate the corresponding discussions in the rebuttal  to elaborate our work and also the potential future directions.
> > > > >
> > > > > Sincerely, Authors

---

### Official Review · Reviewer_sSKR · 2024-07-15

**Soundness:** 2
**Presentation:** 3
**Contribution:** 2
**Rating:** 5
**Confidence:** 3

**Summary:**

In this work, the authors leverage the framework of Kernel Principal Component Analysis (KPCA)
for Out-of-Distribution (OoD) detection, aiming to enhance the separability between In-Distribution
(InD) and OoD data within the subspace defined by the principal components. The study introduces
two task-specific kernels: a cosine kernel and a cosine-Gaussian kernel, both meticulously designed
for OoD detection. To address the computational challenges associated with large-scale kernel
machines, the authors employ Random Fourier Features (RFFs) to approximate the Gaussian kernel
functions. This approach not only maintains the effectiveness of the kernel methods but also
potentially reduces the time complexity of KPCA detectors, making them more feasible for practical
applications.

**Strengths:**

- Innovative Kernel Perspective on Existing KNN Detector: The authors creatively take a kernel perspective on an existing k-nearest neighbor (KNN) detector. This novel approach opens new avenues for improving OoD detection performance by leveraging advanced kernel methods.
- Introduction of Cosine and Cosine-Gaussian Kernels: The proposal of a cosine kernel and a
cosine-Gaussian kernel for Kernel Principal Component Analysis (KPCA) is a significant
contribution. These kernels provide alternative ways to capture non-linear patterns in the data,
enhancing the effectiveness of the OoD detection process.
- Utilization of Explicit Feature Mappings: The paper leverages two explicit feature mappings $\Phi(\cdot)$ induced from the proposed kernels on the original features $z$. This methodical approach showcases the practical application of kernel methods in transforming the feature space for improved detection accuracy.
- Well-Written and Insightful: The paper is well-written, clearly presenting complex concepts and
methodologies. It provides valuable insights into the application of kernel methods for OoD
detection.

**Weaknesses:**

- **Lack of Evidence Against PCA Method:** There is insufficient evidence to substantiate the claim that the failure of the Principal Component Analysis (PCA) method in OOD detection is due to its reliance on linear mapping. Additional experiments and analyses are necessary to support this assertion.
- **Outdated Baselines in OOD Detection:** While the experiments on ImageNet-1K show that the proposed Kernel Principal Component Analysis (KPCA) method, which explores non-linear patterns, is advantageous compared to nearest neighbor searching, it remains uncompetitive within the broader out-of-distribution (OOD) detection community. The baselines used for comparison are outdated, undermining the reliability and relevance of the results.
- **Insufficient Demonstration of $\ell$2-Normalization Significance:** The demonstration of the indispensable significance of $\ell$2-normalization is inadequate. More thorough evidence and experiments are required to convincingly establish its critical role in the proposed methodology.
- **Manual Selection of Kernels and Parameter Tuning**: The specific kernels employed in the study are manually selected and require carefully-tuned parameters. This manual selection process raises concerns about the generalizability and robustness of the proposed method, as it may not perform optimally across diverse datasets and conditions without significant manual intervention.

**Questions:**

- **Evidence Against PCA Method:** The paper mentions the failure of PCA in OoD detection due to its linear mapping. Can you provide more detailed evidence or experiments that explicitly show this limitation of PCA?
- **Extended Experimental Analysis:** Conduct a more comprehensive experimental analysis, including comparisons with the latest state-of-the-art methods in OoD detection. This will help validate the effectiveness and competitiveness of your proposed method in the context of recent advancements in the field.

**Limitations:**

- Manual Selection of Kernels and Parameters: One significant limitation of the proposed method is the manual selection and careful tuning of the cosine and cosine-Gaussian kernels. This process may not generalize well across different datasets and requires extensive manual intervention, which can be time-consuming and prone to human bias.
- Comparative Analysis with Outdated Baselines: The study compares the proposed KPCA-based method with outdated OoD detection baselines. This limits the assessment of the method's competitiveness and effectiveness in the context of recent advancements in the field. Including comparisons with the latest state-of-the-art methods would provide a more accurate evaluation.

---------------------------------------------------------------------------------------
After Rebuttal

The author's detailed response has addressed most of my concerns. I have increased my score. However, I cannot concur with certain aspects, particularly the selection of the kernel and hyperparameter tuning. The FPR cannot serve as a tuning target during training because the OOD datasets are test data. In the setup of this study, only ID data are available for kernel selection and hyperparameter tuning.

---

> ### Author Rebuttal · Authors · 2024-08-06
>
> Thanks for the detailed comments, each of which will be answered pointwisely below.
> Due to length limit, we
> - use "W", "Q", "L" for Weakness, Question, Limitation,
> - put references and mentioned figures in global response and one-page rebuttal PDF file,
> - only show the average FPR in tables and put the results on each OoD dataset in global response.
>
> ---
> ### **(W1 & Q1) Evidence against PCA**
> The primary goal in OoD detection is to differentiate OoD and InD data. To this end, we provide the following evidence on the significance of non-linear KPCA upon linear PCA.
> - As shown in **Fig.1(a)**, InD features from the DNN backbone are not compactly located nor easily separable w.r.t OoD features. Then, it is difficult to rely on  PCA to find a new axis system simply through linear transformation, where InD and OoD features can be well separated, since both InD and OoD features can have worse reconstruction along the axes.
> - In contrast, in **Fig.1(b)** with our KPCA-based method, InD features are located compactly. In this way, InD features can be well reconstructed along the informative axes extracted for InD, while OoD features cannot be well reconstructed with such axes and thus are easily separated from InD.
> - An ablation study is also given in **Table sSKR-1**, further verifying the significance of KPCA over PCA.
>
> **Table sSKR-1** *Comparisons on PCA and KPCA with ResNet18 on CIFAR10 as InD.*
> Method|Average-FPR ($\downarrow$)
> :-:|:-:|
> PCA|66.76
> KPCA w. Cosine|37.78
> KPCA w. Cosine-Gaussian|**27.34**
>
> ---
> ### **(W2 & Q2 & L2) Outdated Baselines**
> As suggested, we additionally compare with four latest detection methods, SCALE [1] (ICLR'2024), DDCS [2] (CVPR'2024), ASH [3] (ICLR'2023) and DML [4] (CVPR'2023). **Table sSKR-2** shows that our method maintains superior performances and will be incorporated into next version.
>
> **Table sSKR-2** *Comparisons on latest baselines with ResNet50 on ImageNet-1K as InD, where our CoRP is set upon ReAct [5,6] similar to Table 3 in the paper.*
> Method|Average-FPR ($\downarrow$)
> :-:|:-:|
> SCALE (ICLR'2024)|20.05
> DDCS (CVPR'2024)|19.36
> ASH-B (ICLR'2023)|22.73
> ASH-S (ICLR'2023)|22.80
> DML+ (CVPR'2023)|29.79
> Ours|**17.68**
>
> ---
> ### **(W3) Insufficient demonstration on $\ell_2$-normalization significance**
> We will elaborate the significance of $\ell_2$-normalization, i.e., Cosine kernel, from two aspects.
> - **Promoting separability between InD and OoD features**
>   - **Motivation.** Kernels can measure (dis)similarities between samples, e.g., a pair of InD features, such that $k(z_1,z_2)=\phi(z_1)^\top \phi(z_2)$. The norms of InD features from the DNN backbone can vary drastically, even after feature mapping $\phi$. This may pose numerical defects to kernel methods, as any two InD features $z_1,z_2$ can have varied magnitudes with very large $\phi(z_1)^\top\phi(z_2)$, despite of their true (dis)similarities. This does not help reveal the intrinsic connections between samples, hindering the extraction of informative principal components of InD features. Then, both InD and OoD features can have bad reconstruction and are thereby not distinguishable.
>   -  **Numerical Tests.** In **Fig.4** and **Fig.5** of **Appendix D.1**, detailed discussions are presented. More specifically,
>        - **Fig.4 (left)** shows the inseparability of InD and OoD features without normalization.
>        - **Fig.4 (middle)** still shows the inseparability of InD and OoD features even applied with a Gaussian kernel, indicating that a non-linear mapping alone fails to capture informative patterns of InD features, nor to separate InD and OoD features.
>        - **Fig.4 (right)**  shows distinctive separability of $\ell_2$-normalized (with Cosine kernel) InD and OoD features, indicating that $\ell_2$-normalization is critical to bring meaningful measures in kernel methods for InD features.
>        - **Ablation studies** are provided in **Table sSKR-3** with thorough results in **Fig.5**.
>
> **Table sSKR-3** *Ablation on Cosine kernel in CoP and CoRP with ResNet18 on CIFAR10 as InD.*
> Method|Average-FPR ($\downarrow$)
> :-:|:-:|
> CoP|**37.78**
> CoP w/o Cosine|66.76
> CoRP|**27.34**
> CoRP w/o Cosine|94.76
> -  **Alleviating imbalanced feature norms for distinguishable reconstruction errors**
> The norms of InD features are commonly larger than that of OoD features, as shown in **Fig.3** in **Appendix D.1**, which was also pointed in [7]. Recall the reconstruction error in Eqn.(4), where $\mu$ is for centering:
> $$e(z)=||U_qU_q^{\top}(z-\mu)-(z-\mu)||_2\leq||U_qU_q^{\top}-I||_2\cdot||z-\mu||_2$$
> As $||z-\mu||_2$ of InD are commonly larger than OoD, _the reconstruction error of OoD features can still be small_, due to the imbalanced norms shown in **Fig.3**, which brings difficulty to differentiate OoD and InD through reconstruction errors. With our deployed $\ell_2$-normalization, the issue of imbalanced norms is greatly alleviated.
>
> ---
> ### **(W4 & L1) Manual kernel selection and parameter tuning**
> The choices of kernels and their tuning parameters are common in kernel methods. Our comprehensive evaluations on OoD detection show:
> - the $\ell_2$-normalization from Cosine kernel plays a pivotal role, where the explicit feature mapping has no hyper-parameters;
> - the $\ell_2$-distance preserving property of the shift invariant Gaussian kernel further promotes separability between InD and OoD, where only $M$ (dimensionality in RFF) and $\gamma$ (band width) need to be tuned. Our practice shows that our default settings of $M=2m$ ($m$ is the InD feature dimension) and $\gamma=1$ with superior performances in most cases can be a mild suggestion for practitioners.
>
> In Sec.6, we have discussed this limitation as mentioned by the Reviewer. Nevertheless, it seems that in practical setups, the kernel choice is relatively simple and does not suffer from exhaustive tuning. We would also like to highlight that our work takes the first step of casting a kernel perspective into OoD detection, advocating more future work.

---

> ### Author Response · Authors · 2024-08-12
>
> Dear Reviewer,
>
> We hope that our responses above could address your concerns and be helpful for final evaluations on our work.
>
> As the author-reviewer discussion is approaching the end, we would like to inquiry  if there is any question, and we would be willing to discuss.
>
> Sincerely, Authors.

---

> > ### Comment · Reviewer_sSKR · 2024-08-13
> > **Thank you for your response.**
> >
> > [W1] Could you provide a complete statement regarding the motivation behind this work? Specifically, addressing two critical questions is essential: 1. Does PCA significantly impact OOD detection? 2. Is the importance of non-linear, low-dimensional structures evident in OOD detection?
> >
> > [W2] Does "Ours" refer to "ReAct+CoRP"?
> >
> > Is your proposed method a boosting scheme? According Table 1, the performance of CoRP appears to be bad.
> >
> > [W3] The proposed method necessitates the selection of a kernel function and bandwidth, with training conducted without access to out-of-distribution (OOD) data. The tuning objective is to maintain a high true positive rate (TPR). However, for any chosen kernel function and bandwidth, a threshold can be determined to achieve a 95% TPR. What is the principle governing hyperparameter selection in your method?

---

> ### Author Response · Authors · 2024-08-13
> **Response to W1**
>
> Dear Reviewer,
>
> Thanks for your kind reply. Below we provide a pointwise response to each of the  further comments.
>
> ---
> ### **(W1) Motivation behind our work**
>
> **(A1)** The motivations from PCA to KPCA are elaborated below.
> - The linear transformations by simply applying PCA [6] cannot capture informative principal components of InD features. As shown in **Fig.1(a)**, InD features are not even located compactly nor linearly-separable with OoD features. In this case, the axes extracted from PCA are insufficient to depict the intrinsic patterns of InD features, so the both InD features and OoD features can have bad reconstructions, hindering to well differentiating OoD from InD.
> - Despite of the current bad performance of PCA, the **low-dimensional subspace** property is commonly considered to boost robustness and thus should be of great interest in OoD detection, as long as informative components can be extracted, such that the intrinsic patterns of InD features are kept, while removing redundant information. Hence, introducing appropriate **non-linear** feature mappings is promising as a remedy to the existing obstacle, motivating us to construct our KPCA-based detector.
> - With KPCA, we can proceed the task in a new **non-linear** feature space, where with proper kernels (new feature spaces), the corresponding InD mappings are located more compactly and even almost linearly separable from OoD mappings, as shown in **Fig.1(b)**. In this mapped space, KPCA can learn informative principal components from InD data which produce distinguishable reconstruction errors w.r.t OoD.
>
> For the raised two questions, we answer below.
>
> **(Q1) Does PCA significantly impact OoD detection?**
> Yes. (K)PCA learns a **low-dimensional** subspace spanned by the (nonlinear) principal components from the InD data. All InD data are utilized to learn such a low-dimensional subspace, while KNN [2] only utilizes the nearest affinity neighborhood in InD data. With the low-dimensional subspace technique, it is expected that the most informative patterns of InD are kept and redundant information in InD is removed, and meanwhile OoD features are easily separable in such space. In this way, based on projections onto the subspace,  the good reconstruction of InD could well differentiate InD and OoD and serve as an effective OoD detection score. Here, the critical problem is how to find an effective low-dimensional subspace in an efficient way, which is well resolved by our proposed KPCA-based detector. **Figure 1** can be referred for illustration.
>
> **(Q2) Is the importance of non-linear, low-dimensional structures evident in OoD detection?**
> Yes. The importance of **low-dimensional structures** has been elaborated in the response to **(Q1)** above. This can also be observed from the improved performance of CoRP than KNN, which all utilize the $\ell_2$ distance/similarities between normalized InD features in **Table 1**. The **non-linearity** lies in the non-linear kernel which is in correspondence to a mapped feature space, elaborated in **(A1)** above. With proper non-linear transformations, significantly improved separability between InD and OoD is attained, as shown by the improved performance of CoP/CoRP than PCA in **Table sSKR-1**.

---

> ### Author Response · Authors · 2024-08-13
> **Response to W2 and W3**
>
> ### **(W2) Our method and the boosting scheme**
>
> Our proposed method is *not* simply a boosting scheme, but a detection method via the KPCA reconstruction error being the detection score.
>
> **The "*bad*" results in Table 1**
> In **Table 1**, CoRP is under a fair comparison with the KNN method [2], where the detection procedures are both straightforwardly proceeded with the DNN features. Besides, the results in Table 1 are not truly bad, since CoRP outperforms the related KNN method in performances and also efficiency as in **Table 2**.  Note that CoRP also achieves better detection than other compared popular baselines, which do not involve boosting techniques.
>
> **The boosting scheme**
> In **Table sSKR-2** of our rebuttal and **Table 3** of Sec.4.2, the boosting scheme is adopted, as we aim to present a fair comparison with the relevant PCA-based method [6]. The PCA method alone cannot achieve reasonable detection performance, unless its detection score gets fused with other methods as shown in [6], e.g., ReAct+PCA. Therefore, we report the results of our KPCA detector *under the same boosting scheme*, e.g., ReAct+CoRP, which refers to the "ours" in **Table sSKR-2** above.
>
> *Remark:* We would like to point out that the strong SOTA methods compared in **Table sSKR-2** and **Table 3** are specifically designed with incorporating boosting schemes, e.g., the "DICE+ReAct" in Table 3 and SCALE [1], DDCS [2] in **Table sSKR-2**, where the feature-pruning-based detectors SCALE [1] and DDCS [2] are boosting detection scores on logits by clipping features. Therefore, in this set of experiments, our boosting implementation is fair for comparisons. Without the boosting technique, our method also achieves superior performances than the compared methods in Sec.4.1.
>
> ---
> ### **(W3) Hyper-parameter selection**
>
> Maintaining a high true positive rate (TPR) and our hyper-parameter tuning principle are elaborated below.
>
> - **Maintaining a 95% TPR** is a common setup in current evaluations on OoD detection methods. Specifically, for any detection method, it is required to achieve a 95% TPR on the InD test set firstly to assign the value of $s$ in Eqn.(1), then, based on the assigned $s$, the detection results on OoD datasets (FPR and AUROC) are determined, which is a widely-adopted procedure in evaluating OoD detection methods.
>
> - **The hyper-parameter selection** is to obtain the best detection FPR values on OoD datasets, in order to find the best performance that can be achieved by the detection method, which is also a common setup in existing works. For example, in prevailing feature-pruning-based OoD detectors [1,2,3,5] with the pruning thresholds as hyper-parameters, their detection results are recorded under a variety of different thresholds and the best results are reported finally, which provides a comprehensive understanding on the feature pruning. Regarding our KPCA detector, we also give a detailed analysis on the effect of each hyper-parameter in our method in **Appendix D.3**, for in-depth insights on the KPCA detector.

---

### Author Rebuttal · Authors · 2024-08-06

Dear Program Chairs, Area Chairs, and Reviewers,

First of all, we would like to thank you for your time, constructive suggestions, which greatly help us improve the work. In this global response, we
- provide full results of tables in the response to Reviewer **sSKR**,
- gather the shared references in the responses to Reviewers **sSKR**, **w1ZW** and **6eWv**,
- put the figures mentioned in responses in the one-page PDF for reference.

---
### **Full results of Tables sSKR-1, sSKR-2 and sSKR-3**

**Table sSKR-1** *Comparisons on PCA and KPCA. The FPR values of each OoD data set are presented with ResNet18 trained on CIFAR10 as InD data.*
Method|SVHN|LSUN|iSUN|Textures|Places365| Average-FPR ($\downarrow$)
:-:|:-:|:-:|:-:|:-:|:-:|:-:|
PCA|30.22|78.15|85.88|46.29|93.27|66.76
KPCA w. Cosine|11.56|23.24|53.71|26.28|74.11|37.78
KPCA w. Cosine-Gaussian|20.68|19.19|21.49|21.61|53.73|**27.34**

---
**Table sSKR-2** *Comparisons on the supplemented recent baseline methods. The FPR values of each OoD data set are presented with ResNet50 trained on ImageNet-1K as InD data. Our CoRP+ detector is set upon ReAct [5,6] and the results can refer to Table 3 in Sec.4.2 of the manuscript.*
method|iNaturalist|SUN|Places|Textures|AVERAGE-FPR ($\downarrow$)
:-:|:-:|:-:|:-:|:-:|:-:|
SCALE (ICLR'2024)|9.50|23.27|34.51|12.93|20.05
DDCS (CVPR'2024)|11.63|18.63|28.78|18.40|19.36
ASH-P (ICLR'2023)|44.57|52.88|61.79|42.06|50.32
ASH-B (ICLR'2023)|14.21|22.08|33.45|21.17|22.73
ASH-S (ICLR'2023)|11.49|27.98|39.78|11.93|22.80
DML (CVPR'2023)|47.32|57.40|61.43|52.80|54.74
DML+ (CVPR'2023)|13.57|30.21|39.06|36.31|29.79
CoRP+ (ours)|10.77|18.70|28.69|12.57| **17.68**

---
**Table sSKR-3** *Ablation on Cosine kernel in CoP and CoRP. The FPR values of each OoD data set are presented with ResNet18 trained on CIFAR10 as InD data.*
Method|SVHN|LSUN|iSUN|Textures|Places365|Average-FPR ($\downarrow$)
:-:|:-:|:-:|:-:|:-:|:-:|:-:|
CoP|11.56|23.24|53.71|26.28|74.11|**37.78**
CoP w/o Cosine|30.22|78.15|85.88|46.29|93.27|66.76
CoRP|20.68|19.19|21.49|21.61|53.73|**27.34**
CoRP w/o Cosine|94.93|95.09|94.53|94.49|94.78|94.76

---
### **References throughout the responses to Reviewers sSKR, w1ZW and 6eWv**

[1] Xu, et al. Scaling for Training Time and Post-hoc Out-of-distribution Detection Enhancement. ICLR'2024.
[2] Yuan, et al. Discriminability-Driven Channel Selection for Out-of-Distribution Detection. CVPR'2024.
[3] Djurisic, et al. Extremely Simple Activation Shaping for Out-of-Distribution Detection. ICLR'2023.
[4] Zhang, et al. Decoupling maxlogit for out-of-distribution detection. CVPR'2023.
[5] Sun, et al. React: Out-of-distribution detection with rectified activations. NeurIPS'2021.
[6] Guan, et al. Revisit pca-based technique for out-of-distribution detection. ICCV'2023.
[7] Sun, et al. Out-of-distribution detection with deep nearest neighbors. ICML'2022.
[8] Pham, et al. Fast and scalable polynomial kernels via explicit feature maps. KDD'2013.
[9] Bakır, et al. Learning to find pre-images. NeurIPS'2004.
[10] Radhakrishnan, et al. Mechanism for feature learning in neural networks and backpropagation-free machine learning models. Science 2024.

---

### Author Response · Authors · 2024-08-14
**Summary of Rebuttal and Discussions**

Dear Chairs and Reviewers,

Thanks for the constructive comments and suggestions during the review process. As the discussion period is approaching the end, we summarize the outcomes below.

---
**Strengths & Reviewers' support:**
1. *Contribution to the kernel perspective for the OoD detection community.*

   All reviewers agree on our contribution of introducing a kernel perspective into OoD detection, which has not been extensively explored in the community.

   Our main contribution lies in the novel kernel perspective on OoD detection with carefully-devised kernels for effectiveness and approximate explicit feature mappings for efficiency,  as recognized by the reviewers on our detection performances and computation efficiency.


2. *Thorough experimental evaluations with strong performances and efficiency.*

   Our method has been  extensively evaluated  with comparisons to many very recent SOTA methods. In rebuttal, as suggested by *Reviewer sSKR* and *Reviewer w1ZW*,  **four very latest methods**, i.e.,  SCALE [1] (ICLR'2024), DDCS [2] (CVPR'2024), ASH [3] (ICLR'2023) and DML [4] (CVPR'2023), have also been added into comparisons in **Table sSKR-2**,  further demonstrating our superior performances.

   Aside of the performances, careful ablation studies have also been presented, as shown in **Table sSKR-3** and **Fig.5**, **Fig.6** in **Appendix D.1** and **Appendix D.2**, covering the imbalanced norms of InD and OoD features, our specified kernel designs, and also comprehensive sensitivity analysis of the hyperparameter as presented in **Appendix D.3**, facilitating more in-depth understandings on our  method.


---
In the rebuttal, we have responded to each of raised comments. *Reviewer 6eWv and Reviewer LuQe recognized that our responses have addressed their questions during the discussions; Reviewer LuQe mentioned to vote to accept our work and Reviewer 6eWv increased the score.*

**Remaining comments in the review:**
Below, we summarize the remaining comments which we may have no time further discuss about.

> Reviewer sSKR: Could you provide a complete statement regarding the motivation, and explanations on the boosting scheme and the hyper-parameter selection?

We have elucidated the motivation from PCA to KPCA in the latest rebuttal of **Response to W1**, which mainly lies in:
- the advantage of the **low-dimensional** subspace learned by (K)PCA in boosting robustness;
- the **non-linearity** of kernels that improves the limitation of PCA in inseparable cases.

Besides, in the rebuttal of **Response to W2 and W3**, we have also clarify the relationship between our method and the boosting scheme, and the principal governing the hyper-parameter selection.

> Reviewer w1ZW: The studied problem is very important. The results seem to outperform state-of-the-art.
> Reviewer w1ZW also raised suggestions to add comparisons with more methods and kernel designs, elaborations on the significance of linear separable space, and our assumptions.

We thank the *Reviewer w1ZW*'s acknowledgement on the significance and effectiveness of our work. We have responded to each comment with both empirical evidence and explanations. By far, we have not received the feedback from the reviewer, though we would be very willing to discuss and see if any further advice would be received.

---
**Updates to the manuscript:**

We thank all reviewers for the suggestions in the review and discussion periods. Accordingly, the new experimental results and the more clarified elaboration on our work in our responses will be added to update the manuscript, which we believe are durable in the current status. We hope that our summaries above and pointwise responses below will be helpful for the final decision of our work.

Best wishes,
Authors of Paper 5729

[1] Xu, et al. Scaling for Training Time and Post-hoc Out-of-distribution Detection Enhancement. ICLR'2024.
[2] Yuan, et al. Discriminability-Driven Channel Selection for Out-of-Distribution Detection. CVPR'2024.
[3] Djurisic, et al. Extremely Simple Activation Shaping for Out-of-Distribution Detection. ICLR'2023.
[4] Zhang, et al. Decoupling maxlogit for out-of-distribution detection. CVPR'2023.

---

### Public Comment · ~Vangjush_Kostandin_Komini1 · 2025-05-13
**Explanation about Mahalanobis distance in equation 12**

Regarding the Mahalanobis distance in equation 12. Is it possible that the covariance matrix should be $\Sigma^{-1}_{l}$ ?

---

> ### Public Comment · ~Kun_Fang1 · 2025-05-14
>
> Thank you for catching this issue. The pseudo-inverse is indeed missing in Eqn.(12). Apologies for this typo.

---

### Decision · Program_Chairs · 2024-09-25

**Decision:**

Accept (poster)

**Comment:**

This paper proposes to use a kernel PCA for OOD with deep neural net features. As a choice of kernel, this study suggests to use a cosine kernel and a cosine-Gaussian kernel for the OOD tasks. In addition to that, the usage of Random Fourier Features (RFFs) is also suggested to reduce the computational complexity. The practical effectiveness of the proposed method is examined through several numerical experiments.

Although the usage of kernel PCA is not very novel, the authors enhanced their proposal by several numerical experiments. In addition to the main submission, they conducted some more ablation studies in the rebuttal to address reviewers' concerns.
The authors experimentally demonstrated the effectiveness of the proposed method against the existing nearest neighbor method, which would be informative to the community.

This paper was quite on the borderline, but based on the discussions with the reviewers after rebuttal, I would like to recommend acceptance of this paper.